



# Evaluation of "on site" calibration procedures for sun-sky photometers

Monica Campanelli [1], Victor Estelles[2], Gaurav Kumar [2], Teruyuki Nakajima[3], Masahiro Momoi [4],
Julian Gröbner[5], Stelios Kazadzis[5], Natalia Kouremeti[5], Angelos Karanikolas[5], Africa Barreto[6],
Saulius Nevas[7], Kerstin Schwind[7], Philipp Schneider[7], Iiro Harju[8], Petri Kärhä[8], Henri Diémoz[9],
Rei Kudo[10], Akihiro Uchiyama[11], Akihiro Yamazaki[10], Anna Maria Iannarelli[12], Gabriele Mevi[12],
Annalisa Di Bernardino[13], Stefano Casadio[12].

Institute of Atmospheric Sciences and Climate , CNR, Rome, Italy
University of Valencia, Valencia, Spain
National Institute for Environmental Studies, 16–2 Onogawa, Tsukuba, Ibaraki 305-8506, Japan
GRASP SAS: Generalized Retrieval of Atmosphere and Surface Properties, Villeneuve d'Ascq, France.
Phys. and Meteo. Obs. Davos / World Radiation Center (PMOD/WRC), Davos, Switzerland
6Izaña Atmospheric Research Center (IARC), Agencia Estatal de Meteorología (AEMET), Santa Cruz de Tenerife,
Spain
Physikalisch-Technische Bundesanstalt (PTB), Braunschweig, Germany
Aalto University, Aalto, Finland
ARPA Valle d'Aosta, Saint Christophe, Italy
Meteorological Research Institute, Japan Meteorological Agency, Tsukuba, 305-0052, Japan
National Institute for Environmental Studies, Tsukuba, 305-0053, Japan
SERCO Italy, spa, Frascati, Italy.
Sapienza University of Rome, Department of Physics, Rome, Italy

**Corresponding author:** Monica Campanelli m.campanelli@isac.cnr.it

**Abstract**
To retrieve columnar aerosol properties from sun-photometers both irradiance and radiance calibration factors are needed. For the irradiance the solar calibration constant, $V_0$, that is the instrument counts for a direct normal solar flux extrapolated to the top of the atmosphere, must be determined. The solid view angle, SVA, is a measure of the field of view of the instrument, and it is important for obtaining the Radiance from sky diffuse irradiance measurements. Each of the three sun-photometers networks considered in the present study (SKYNET, AERONET, WMO-GAW) adopts different protocols of calibration, and we evaluated the performance of the on-site calibration procedures, applied to SKYNET PREDE-POM instruments, during intercomparison campaigns and laboratory calibrations held in the framework of the Metrology for Aerosol Optical Properties (MAPP) EMPIR project. The on-site calibration, performed as frequently as possible (rather monthly) to monitor change of the devise condition, allow operators to track and evaluate the calibration status on a continuous basis considerably reducing the data gaps incurred by the periodical shipments for performing centralized calibrations. The performance of the on-site calibration procedures for $V_0$ was very good in sites with low turbidity, showing an agreement with a reference calibration between 0.5% and 1.5% depending on wavelengths. In the urban area, the agreement decreases between 1.7% and 2.5%. For the SVA the difference varied from a minimum of 0.03% to a maximum of 3.46%.

## 1. Introduction

The ground-based remote sensing measurements of the solar radiation are an important part of atmospheric physics aimed to determine the columnar aerosol optical properties. Sun-sky photometers and sun-photometers are instruments performing direct and diffuse solar radiation measurements in the wavelength regions where gases' absorption is low or negligible. Several networks have been established worldwide, such as AERONET (Holben et al., 1998), WMO-GAW (Kazadzis et al., 2018a) and SKYNET (Nakajima et al., 2020). These networks provide well tracked, but with different basic principles, calibration procedures, good quality standards and homogeneity on the retrievals. Traceability and data quality are essential requirements by the World Meteorological Organization (WMO) for monitoring atmospheric aerosol optical properties. In 2006, the Commission for Instruments and Methods of Observation (CIMO) of the WMO (WMO, 2007) recommended that the World optical depth research and calibration center (WORCC) at the PMOD-WRC is designated as the primary WMO Reference Centre for aerosol optical depth (AOD) measurements (WMO, 2005). Since 2000, reference instruments from different networks are intercompared in order to ensure worldwide aerosol optical depth homogeneity (e.g. Kazadzis et al., 2018b, Kim et al., 2008, WMO, 2023).

To obtain columnar aerosol properties from sun-photometers, both irradiance and radiance calibration factors are needed. For the irradiance, the solar calibration constant ($V_0$) must be determined whereas the solid view angle (SVA) is an intermediate step for the radiance calibration. $V_0$ is the instrument counts for a direct normal solar flux, F, (Irradiance, instrument units) extrapolated to the top of the atmosphere (Shaw, 1976), and it is an important issue for the estimation




of the AOD. An error of 10% in the estimation of $V_0$ induces an uncertainty in the retrieval of AOD of about 0.1, therefore
a good accuracy is needed in its determination. SVA is a measure of the field of view of the instrument, and it is important
for obtaining the Radiance, L, $(Wm^{-1}sr^{-1})$ from sky diffuse irradiance measurements (E), being L the ratio between E and
SVA.
Each of the three networks considered in the present study adopts different protocols of calibration. For the AERONET
(Giles et al., 2019) CIMEL sun-sky photometers, $V_0$ is transferred from a value of reference instrument which is retrieved
by Langley-plot based on measurements at a mountaintop calibration site (Shaw, 1976; Holben et al., 1998). The primary
mountaintop calibration sites in AERONET are located at the Mauna Loa Observatory (latitude 19.536, longitude
−155.576, 3402 m) on the island of Big Island (Hawaii) and the Izana Observatory (latitude 28.309, longitude −16.499,
2401 m) on the island of Tenerife in the Canary Islands (Toledano et al., 2018, Cuevas et al., 2022). These reference
instruments are routinely monitored for stability and typically recalibrated every 3 to 8 months. Langley-calibrated
instruments move to main calibration locations (such as Washington DC (USA), the Observatoire de Haute-Provence
(OHP, France) or Valladolid (Spain)), and transfer their calibration to reference instrumentation. Then each of the CIMEL
network instruments are visiting these locations and they are calibrated. Radiance L is directly obtained by a calibration
with the integrating spheres at the AERONET calibration centers, providing an absolute calibration traceable to a NIST
standard lamp hosted at the NASA GSFC calibration facility.
WMO-GAW uses PFR sun-photometers measuring only the direct solar Irradiance. $V_0$ is calculated by comparison
against three Langley-calibrated instruments (triad) at the WORCC (Kazadzis et al., 2018a). The triad is also checked by
comparisons with instruments visiting WORCC every six months operating at Mauna Loa and Izana and perform Langley
calibrations. Within the ACTRIS European research infrastructure, three reference PFRs are permanently located at the
AERONET Europe calibration locations of OHP, Valladolid and Izana to ensure data homogeneity.
SKYNET adopts on-site calibration routines for the PREDE-POMs sun-sky photometers to determine the $V_0$ and SVA,
using the improved Langley plot method described in section 3.3 and the disk scan method (Nakajima et al., 1996; Boi et
al., 1999; Uchiyama et al., 2018) described in section 4.3. The on-site calibration procedures are performed as frequently
as possible (rather monthly) to monitor change of the device condition, since the deterioration of the optical filters or
other parts of the optics is detectable in a change of the temporal behavior of the calibration constants. On-site calibration
procedures allow operators to track and evaluate the calibration status on a continuous basis considerably reducing the
data gaps incurred by the periodical shipments for performing centralized calibrations. Also, the likelihood of instrumental
damages attributable to transport decreases.
In the present work we evaluate the performance of the on-site calibration procedures applied to two PREDE-POM
instruments, using intercomparison campaigns and laboratory calibrations held in the framework of the Metrology for
Aerosol Optical Properties (MAPP) EMPIR project. The overall aim of MAPP is to enable the SI-traceable measurement
of column-integrated aerosol optical properties retrieved from the passive remote sensing of the atmosphere using solar
and lunar radiation measurements.
**2. Sites and instruments**
The on-site calibration procedures were applied to four different PREDE POMs of the SKYNET network (Table 1), using
datasets from the campaigns held in two mountain sites, Davos (9.846W, 46.814N, 1588.4 m a.s.l), and Izana (16.499E,
28.309N, 2373.0 m a.s.l), and in two urban sites, Rome (12.516W, 41.902N, 83.0, m a.s.l) and Valencia (0.418E,
39.508N, 60.0m a.s.l). The periods of the campaigns are also listed in Table 1:
Table1: List of the campaigns used for the evaluation of the on-site calibration procedure performance; * POM_CNR is
a Lunar and solar version.

| Campaign name | Location | Involved Instr. | Period |
|---|---|---|---|
| QUATRAM 1 | Davos | POM_VDV | 10/08/2017-31/08/2017 |
| QUATRAM 1 | Rome | POM_VDV | 22/09/2017-11/03/2017 |
| QUATRAM 2 | Davos | POM_22 | 24/07/2018-19/10/2018 |
| QUATRAM 2 | Rome | POM_22 | 01/05/2019-30/09/2019 |
| MAPP-QUATRAM 3 | Rome | POM_CNR * | 03/09/2021-20/09/2021 |
| FRC-QUATRAM 3 | Davos | POM_CNR* | 07/10/2021-19/10/2021 |
| MAPP Valencia | Valencia | POM_UV | 04/10/2022-30/11/2022 |
| MAPP Izana | Izana | POM_CNR* | 02/09/2022-22/09/2022 |

The QUAlity and TRaceabiliy of Atmospheric aerosol Measurements (QUATRAM) campaigns (Campanelli et al., 2018;
http://www.euroskyrad.net/quatram.html) are organized by the Institute of Atmospheric Science of CNR (Italy) and the
Physikalisch-Meteorologisches Observatorium Davos/World Radiation Center (PMOD/WRC). They are aimed to evaluate
the homogeneity and comparability among measurements performed by equipment of different International Networks
and/or manufactures, and to assess the accuracy of the new on-site calibration procedures. The Networks/Instruments
involved in QUATRAM are: SKYNET-PREDE/POM sun-sky photometers; AERONET-CIMEL 318 photometers;
WMO-Precision Filter Radiometer (PFR); Multi Filter Rotating Shadowband Radiometers (MFRSR) and Middleton
photometers. The approach consists of performing a calibration transfer from a primary master PFR of the PMOD/WRC



to the other instrumentation, of the comparison of AODs at the common wavelengths, and of the evaluation of the on-site
calibration procedures. The campaigns were held in both urban (Rome) and mountain (Davos) sites to consider different
atmospheric turbidity and aerosol optical characteristics. The QUATRAM 3, held in Davos in 2021, was hosted by the
Fifth WMO Filter Radiometer Comparison (FRC-V) (WMO, 2023).
The Izana and Valencia campaigns were held in the framework of the Metrology for aerosol optical properties (MAPP)
project with the purpose of generating data to be used for a development of a comprehensive uncertainty budget for
aerosol optical properties from remote sensing techniques and to determine the Top-of-Atmosphere solar and lunar
spectra.
The evaluation of the performance of the SKYNET on-site calibration procedures was assessed by comparing the
retrieved constants against:
a.  the laboratory calibrations performed by the Physikalisch-Technische Bundesanstalt (PTB), Germany, the Aalto
124          University, Finland, and the PMOD, Switzerland.
b.  the transfer of calibration from PFR and CIMEL to PREDE-POM instruments operating simultaneously.
The PFR instrument, manufactured by PMOD/WRC, is used in the GAW AOD network, and it is a classic sun photometer
equipped with 3 to 5 nm bandwidth interference filters (368nm, 412 nm, 500 nm, 863 nm) and a field of view of 2.5∘.
The detector unit is held at a constant temperature of 20 ∘C by an active Peltier system. Dielectric interference filters
manufactured by the ion-assisted deposition technique are used to assure significantly larger stability in comparison to
manufactured by classic soft coatings. The PFR was designed for long-term stable measurements; therefore, the
instrument is hermetically sealed with an internal atmosphere that is slightly pressurized (2000 hPa) with dry nitrogen.
The CIMEL CE 318, standard AERONET instrument (Holben et al., 1998; Giles et al., 2019), is a multi-wavelength
automatic sun-sky photometer developed by Cimel Electronique, measuring direct solar irradiance and sky radiance at
nine bands (340 nm, 380 nm, 440 nm, 500 nm, 675 nm, 870 nm, 937 nm, 1020 nm, and 1640 nm) with 2-10 nm Full
Width at Half Maximum (FWHM) and a field of view of 1.3° (Torres et al., 2013). The detector is not thermostated and
corrections are performed a-posteriori. The PREDE-POM, standard instrument of the SKYNET network, is a sun-sky
photometer operating at seven wavelengths in the model 01 (315 nm, 400 nm, 500 nm, 675 nm, 870 nm, 940 nm, 1020
nm). Three of the four PREDE-POMs however have been modified by replacing the 315 nm filter with a filter at 340 nm.
The field of view is 1° and FWHM is equal to 3 nm (UV) and 10 nm (visible, VIS and near-infrared). The optics are
thermostated at 30°C.
**3.  Estimation of the Solar calibration constant**
Six methods for the estimation of $V_0$ are analysed in the following sections: the in-lab calibration at PTB, the transfer of
calibration among instruments and the on-site procedures.
**3.1 The laboratory calibrations at PTB**
The two sun-sky radiometers, POM_UV and POM_CNR, were calibrated at PTB with respect to their spectral irradiance
responsivities. The calibrations were accomplished using the tunable laser-based facility, TUable Lasers In Photometry
(TULIP). The TULIP facility, shown in Figure 1, has recently been upgraded with a laser system based on an optical
parametric oscillator (OPO) operating in pulsed mode with a pulse length of 2.5 ps and a repetition rate of 80 MHz. The
laser wavelength is automatically tunable throughout the spectral range from 230 nm to 2300 nm. A high-accuracy laser
spectrum analyzer (LSA) is used to monitor the laser wavelength, which is stable within 10 pm during a typical
measurement sequence. The spectral bandwidth of the laser radiation is wavelength-dependent and varies between 0.2
nm and 0.7 nm in the visible spectral range. The centroid values of the measured laser spectrum are used as the
wavelengths of the corresponding spectral responsivity values.
A spatially homogeneous non-polarized field with temporally stabilized irradiance values is produced by a beam shaping
optics based on a micro lens array. The amplitude stabilization of the output radiation from the laser system is achieved
using two liquid crystal display (LCD)-based modulators inserted in the signal and idler beams of the OPO, before the
second and third harmonic (SHG and THG) modules of the laser system. The feedback signals for the control circuits of
the intensity modulators are taken from Si and InGaAs photodiodes irradiated by a fraction of the radiation field formed
by the micro lens array. In this way, the irradiance values at the measurement plane are stabilized to a level of a few parts
in $10^4$. The homogeneity of the generated field is within a few parts in $10^3$. Spectral irradiance responsivity calibrations
are made in such a field by comparing the signal of a device under test (DUT) to that of a reference detector (REF),
positioned sequentially at the same position in the measurement plane. The spectral irradiance responsivities of the
reference detectors built of Si and InGaAs photodiodes for the visible and near infrared wavelengths, respectively, are
obtained through a chain of calibrations from a primary cryogenic radiometer and from the calibrated areas of the precision
radiometric apertures used with the reference detectors.
The spectral irradiance responsivity calibrations of the sun photometers were made at ca. 1.5 m from the micro lens array.
At this distance, the illuminated area of the micro lens array seen by the radiometers subtends ca 0.3 degrees. The entrance
apertures of the sun photometers were aligned perpendicular to the optical axis of the TULIP setup. The angular
orientation of the POM instruments in the setup was optimised by tilting and rotating to maximize the signal. This ensured
that the central part of the field of view was illuminated by the laser-induced irradiation field. The digital signals (DN)
from the POM instruments were requested and read via a serial port of the TULIP control PC using respective software
commands. During the measurements it was not possible to select the internal gain settings of the POMs. These settings





are managed by the instrument firmware. It was therefore also not possible to verify the gain values during the laboratory
calibrations and their respective contributions to the measurement uncertainties.


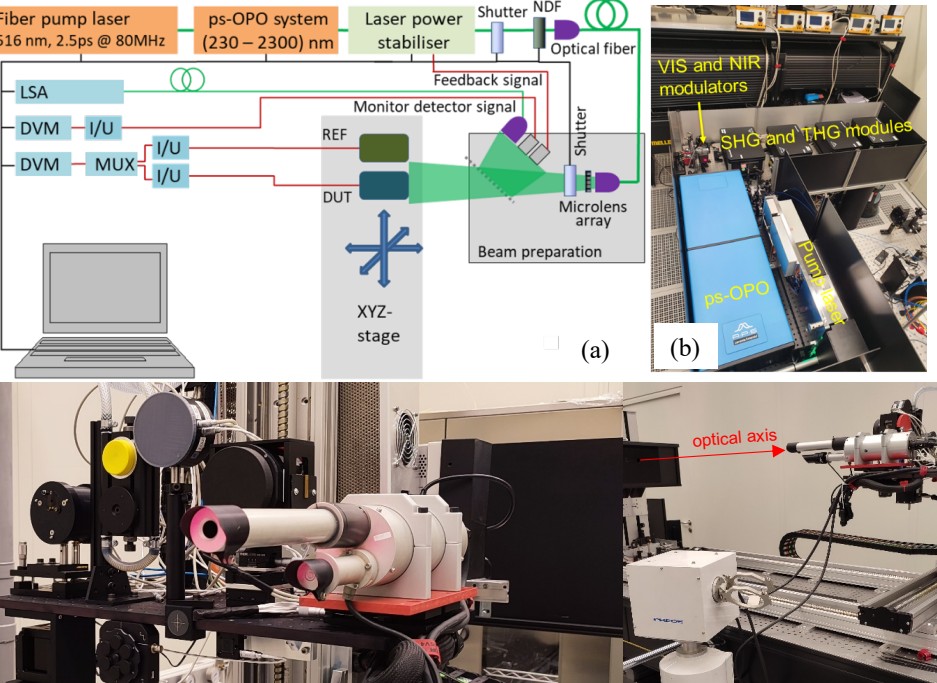

Figure 1. TULIP setup at PTB: (a) schematic representation of the setup including optical parametric oscillator (OPO)
system, variable neutral-density filter (NDF), reference (REF) and detector under test (DUT), current-to-voltage converter
(I/U), multiplexer (MUX), digital voltage meter (DVM) and laser spectrum analyzer (LSA); (b) a picture of the ps-OPO
system; (c) a picture of POM and reference detectors installed on the translation stage system; (d) a side view of the POM
facing the beam shaping optics inside the enclosure.
The results of the calibrations of all the channels of the two instruments are shown in Figure 2. The bandpass functions
of the spectral channels were found to match well the nominal filter function. Only the 940 nm channel of POM_UV
showed a large deviation. Most of the spectral channels were confirmed to block the out-of-band radiation to the level of
1E-8 throughout the whole spectral range.





Figure 2. Measured spectral irradiance responsivities of all channels of the sun photometers and their normalized values displayed on a logarithmic scale.

The uncertainty analysis of the spectral irradiance measurements was accomplished by a Monte Carlo method according to Supplement 1 to the "Guide to the expression of uncertainty in measurement" using the measurement equation including all relevant uncertainty contributions. The know uncertainty components include the uncertainty of the reference detector responsivity, its aperture area, stability and LSA-based measurement of the laser wavelength, spatial homogeneity of the laser-generated field, the temporal stability of the irradiance values, laser bandwidth variation, and positioning of the detectors in the plane of measurements. For the latter uncertainty contribution, the position of the effective radiometric aperture of the measured detectors along the optical axis must be known. In the case of the reference





detectors with well-defined mechanical apertures, their position can be determined with an accuracy of better than 0.1
mm. However, the position of radiometrically limiting apertures of sun photometers with lens optics cannot be measured
directly as they are behind the lens. In this case, they were determined through distance variation with much higher
resulting uncertainties. For the Prede POM sun photometers, the positions of the effective apertures could be determined
with estimated standard uncertainties of 3 mm. The respective uncertainty contribution was also dominating the
uncertainty of the spectral irradiance responsivity calibrations of the filter radiometers (Figure 3).
It should be noted that the uncertainty analysis only included the uncertainty components identified during the laboratory
calibrations under the respective measurement conditions. As mentioned above, uncertainty contributions from internal
gain values of the POMs could not be estimated due to the lack of functionality of the instruments for laboratory
calibrations. Also, the temperature stabilization of the POM_CNR did not work during the calibrations at PTB. The effect
of the instrument malfunction on the calibrated responsivity values was not included in the uncertainty analysis. In
addition, there may be some other differences between the operating conditions of the instruments during the laboratory
calibrations and their use in the field, which could lead to additional uncertainty contributions.

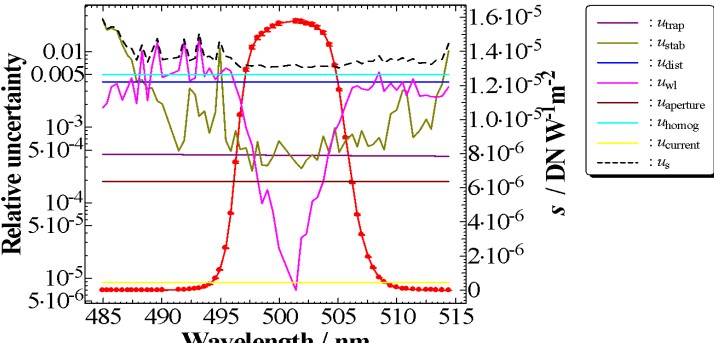

Figure 3. Example of spectrally dependent uncertainty components of spectral irradiance responsivity measurements of
the 500 nm channel of POM_CNR. The relative uncertainties on the left axis represent components due to reference
detector ($u_{trap}$), temporal irradiance stability ($u_{stab}$), detector positioning ($u_{dist}$), laser wavelength ($u_{wl}$), REF aperture area
($u_{aperture}$), spatial homogeneity ($u_{homog}$), photocurrent measurements of REF ($u_{current}$), and the resulting standard uncertainty
of the measurements ($u_s$).
The calibration factors $V_0$ were obtained a posteriori, by integration of the spectral response and the extra-terrestrial TSIS
spectrum (Coddington et al., 2023). The uncertainties were estimated by quadratic error propagation of the numerical
integral. The results are summarised in Table 2.
Within the EMPIR project 19ENV04 MAPP, sun photometers from GAW-PFR and AERONET networks  were also
measured at PTB with respect to their spectral irradiance responsivities. The results of the laser-based calibrations of
several sun photometers were verified by additional methodologies for laboratory calibrations. The spectral irradiance
responsivities of a PFR and two CIMELs determined at the TULIP setup were verified by a calibration against reference
standard lamps with traceability to the primary spectral irradiance standard (a high-temperature blackbody). The results
agreed well within the uncertainties of the calibrations, i.e. in the range between 0,2%  and 1%. One CIMEL was also
calibrated in radiance mode using an integrating sphere source calibrated at PTB for the spectral radiance. This calibration
data combined with the FOV values measured by PMOD yielded spectral irradiance responsivities of the CIMEL channels
that agreed within 1% to 2% to those determined at the TULIP setup in irradiance mode.
The spectral irradiance responsivities of the PRF were combined with the published spectral irradiance at the top of the
atmosphere (TOA) values (QASUMEFTS ($\lambda \leq 500$) & TSIS-1 HSRS ($\lambda > 500$ nm)) to derive the signal values that would
be measured at the TOA. Those values were compared with those obtained by the Langley technique. The agreement
between the values was within 0,5%. Also the AOD values derived using the laboratory-based calibration of the PFR
were well in agreement to those from the Langley-based calibration (Kouremeti et al, 2021; Gröbner et al., 2023).
For the three CIMEL instruments calibrated at PTB, the agreement between the calculated TOA values and those derived
by the Langley extrapolation technique was in the range of 1% to 5%, with the discrepancies systematically increasing
towards the short-wavelength channels. Thus, for all instruments, the results of the in-lab calibrations were consistent
within their respective uncertainties, regardless of the calibration methods used.
### 3.2   The standard Langley (SL) method for POM_CNR
The Standard Langley method (Shaw, 1976) is the most common procedure adopted to calculate the solar calibration
constant. It is based on the Beer-Lambert law (Eq. 1)

Atmospheric
Measurement
Techniques

Discussions


$$V = V_0 \exp(-m_0\tau)$$
$$\text{Or}$$
$$\ln V = \ln V_0 - m_0(\tau_{gas} + \tau_R) - m_0\tau_{ext}$$

Eq.1


Where V is the direct solar irradiance measured at ground, $m_0$ is the optical air mass as the inverse of the
solar zenith angle, $\tau_{ext}$ is the extinction AOD, $\tau_{gas}$ and $\tau_R$ are are respectively the gas absorption optical depth and the
molecular (Rayleigh) scattering optical depth.
The Standard Langley method consists of the retrieval of $V_0$ by the fit of y vs x in Eq. 2, assuming that optical depth due
to aerosol is constant, as it happens performing the measurements at high altitude (i.e. above the boundary layer, where
AOD is low and its absolute variability is also very low).

$$y = a_{SL} + b_{SL}x \quad where$$
$$x = m_0$$
$$y = \ln V + m_0 \cdot (\tau_{gas} + \tau_R)$$

Eq.2


The linear fitting provides intercept $a_{SL} = \ln V_0$ and slope $b_{SL} = -\tau$.
This method is used for measurements taken at the Izaña observatory by the POM_CNR. The following criteria are used
to filter the data: i) only data for $m_0 >= 2$ and $<= 5$ are considered; ii) using $a$ and $b$ parameters retrieved from the fit, $y_{fit}$ is
obtained from Eq.2 and the residuals are calculated for each point as $y - y_{fit}$; their RMSD is calculated and if it is $> 0.006$,
the mean of residuals is calculated and points for which residual is greater than mean value are removed; a new fit is then
performed and the process is repeated until RMSD $< 0.006$ is obtained; iii) special criterion is applied for 340 nm where
data points were only selected for m $< 2$. The primary reason for choosing this airmass threshold is its sensitivity towards
molecules (Rayleigh scattering). Selecting higher optical mass means light gets scattered more and can cause errors. A
similar strategy is also used in Estelles, et al. (2004). The selected series were considered only if the number of data points
are greater than 50. After a visual inspection, three days of the Izana campaign (7, 8 and 9 September, 2022) were very
stable and showed minor fluctuations. Calibration values were calculated for these three days, both in the morning (before
13 UTC) and afternoon for each wavelength with the air mass limit between 2 and 5.
Uncertainty was determined as the standard deviation of the calibration values calculated for three days in morning and
evening (6 plots). The mean was taken as the final calibration value. Results are shown in Table 2a and b.

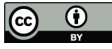



| Location | yymm | POM_CNR | V0_IL*e-04 (A) 340 | 400 | 500 | 675 | 870 | 1020 | %CV 340 | 400 | 500 | 675 | 870 | 1020 | Unc 340 | 400 | 500 | 675 | 870 | 1020 |
|---|---|---|---|---|---|---|---|---|---|---|---|---|---|---|---|---|---|---|---|---|
| Davos | 1708 | IL | | 1.363 | 2.828 | 3.486 | 2.229 | 1.164 | | 2.63 | 1.87 | 1.24 | 1.04 | 1.28 | | 3.5847E-06 | 5.2884E-06 | 4.3226E-06 | 2.3182E-0 | 1.4899E-06 |
| | | XIL | | 1.3411 | 2.8384 | 3.5466 | 2.2336 | 1.1984 | | 4.06 | 2.41 | 3.05 | 1.75 | 4.32 | | 5.4430E-06 | 6.8334E-06 | 1.0821E-05 | 3.8981E-06 | 5.1719E-06 |
| | | PFR | | | 2.844 | | 2.231 | | | | 0.22 | | 0.27 | | | | 6.126E-07 | | 5.933E-07 | |
| Rome | 1710 | IL | | 1.307 | 2.782 | 3.454 | 2.204 | 1.151 | | 2.2 | 1.29 | 0.71 | 0.85 | 1.48 | | 2.8754E-06 | 3.5888E-06 | 2.4523E-06 | 1.8734E-06 | 1.7035E-06 |
| | | XIL | | 1.3101 | 2.7803 | 3.4634 | 2.2171 | 1.1417 | | 6.05 | 2.16 | 1.31 | 0.81 | 1.16 | | 7.9275E-06 | 6.0121E-06 | 4.5426E-06 | 1.7925E-06 | 1.3215E-06 |
| | | PFR | | | 2.858 | | 2.226 | | | | 0.11 | | 0.19 | | | | 3.043E-07 | | 4.266E-07 | |
| Davos | 1807 | IL | 0.0886 | 1.289 | 2.751 | 3.268 | 2.3 | 1.236 | 0.77 | 0.71 | 0.35 | 0.19 | 0.36 | 1.01 | 6.8245E-08 | 9.1519E-07 | 9.6285E-07 | 6.2092E-07 | 8.28E-07 | 1.2484E-06 |
| | | XIL | 0.0896 | 1.3061 | 2.7756 | 3.284 | 2.3203 | | 1.65 | 1.53 | 1.25 | 0.73 | 0.68 | | 1.4809E-07 | 2.0023E-06 | 3.4809E-06 | 2.3990E-06 | 1.5773E-06 | |
| | | PFR | | | 2.781 | | 2.325 | | | | 0.25 | | 0.18 | | | | 6.948E-07 | | 4.221E-07 | |
| Davos | 1808 | IL | 0.0888 | 1.3 | 2.762 | 3.294 | 2.321 | 1.228 | 1.4 | 1.3 | 0.97 | 0.85 | 0.97 | 0.52 | 1.2438E-07 | 1.6900E-06 | 2.6791E-06 | 2.7999E-06 | 2.2514E-06 | 6.3856E-07 |
| | | XIL | 0.0889 | 1.3045 | 2.788 | 3.3126 | 2.3282 | 1.2396 | 3.35 | 2.84 | 2.24 | 1.29 | 0.87 | 0.86 | 2.9799E-07 | 3.7027E-06 | 6.2579E-06 | 4.2749E-06 | 2.0167E-06 | 1.0606E-06 |
| | | PFR | | | 2.799 | | 2.349 | | | | 0.50 | | 0.94 | | | | 1.410E-0 | | 2.204E-06 | |
| Davos | 1809 | IL | 0.0888 | 1.298 | 2.771 | 3.312 | 2.343 | 1.24 | 1.37 | 0.74 | 0.41 | 0.18 | 0.2 | 0.6 | 1.217E-07 | 9.6052E-07 | 1.1361E-06 | 5.9616E-07 | 4.686E-07 | 7.44E-07 |
| | | XIL | 0.0897 | 1.3065 | 2.7897 | 3.3606 | 2.3744 | 1.2379 | 3.42 | 1.68 | 1.52 | 2.17 | 1.18 | 0.50 | 3.0721E-07 | 2.1913E-06 | 4.2431E-06 | 7.2794E-06 | 2.7935E-06 | 6.2217E-07 |
| | | PFR | | | 2.801 | | 2.368 | | | | 0.19 | | 0.43 | | | | 5.387E-07 | | 1.017E-06 | |
| Davos | 1810 | IL | 0.0881 | 1.286 | 2.766 | 3.317 | 2.346 | 1.259 | 0.73 | 0.53 | 0.31 | 0.1 | 0.36 | 0.84 | 6.4298E-08 | 6.8158E-07 | 8.5746E-07 | 3.317E-07 | 8.4456E-07 | 1.0576E-06 |
| | | XIL | 0.0892 | 1.3025 | 2.7791 | 3.3306 | 2.3561 | | 1.01 | 1.09 | 0.54 | 0.42 | 0.24 | | 8.9976E-08 | 1.4159E-06 | 1.4910E-06 | 1.3825E-06 | 5.7300E-07 | |
| | | PFR | | | 2.802 | | 2.364 | | | | 0.14 | | 0.17 | | | | 3.842E-07 | | 3.991E-07 | |
| Rome | 1905 | IL | 0.0857 | 1.274 | 2.717 | 3.268 | 2.321 | 1.235 | 1.23 | 1.1 | 0.55 | 0.54 | 1.81 | 1.17 | 1.0542E-07 | 1.4014E-06 | 1.4944E-06 | 1.7647E-06 | 4.201E-06 | 1.445E-06 |
| | | XIL | 0.0877 | 1.3061 | 2.7466 | 3.2535 | 2.2858 | 1.2063 | 3.13 | 4.75 | 1.32 | 2.29 | 3.48 | 2.23 | 2.7478E-07 | 6.2097E-06 | 3.6318E-06 | 7.4567E-06 | 7.9489E-06 | 2.6940E-06 |
| | | PFR | | | 2.804 | | 2.348 | | | | 0.55 | | 0.55 | | | | 1.552E-06 | | 1.280E-06 | |
| Rome | 1906 | IL | 0.0852 | 1.269 | 2.73 | 3.272 | 2.303 | 1.228 | 0.51 | 0.66 | 0.72 | 0.69 | 0.7 | 0.48 | 4.3467E-08 | 8.3754E-07 | 1.9656E-06 | 2.2577E-06 | 1.6121E-06 | 5.8944E-07 |
| | | XIL | 0.0865 | 1.2875 | 2.7762 | 3.3197 | 2.3329 | 1.2497 | 4.56 | 2.15 | 2.17 | 2.33 | 2.00 | 2.64 | 3.9447E-07 | 2.7702E-06 | 6.0374E-06 | 7.7213E-06 | 4.6677E-06 | 3.3028E-06 |
| | | PFR | | | 2.809 | | 2.347 | | | | 0.68 | | 0.49 | | | | 1.909E-06 | | 1.144E-06 | |
| Rome | 1907 | IL | 0.0841 | 1.261 | 2.737 | 3.257 | 2.299 | 1.231 | 1.67 | 1.61 | 1.07 | 0.47 | 0.37 | 0.26 | 1.4043E-07 | 2.0302E-06 | 2.9286E-06 | 1.5308E-06 | 8.5063E-07 | 3.2006E-07 |
| | | XIL | 0.0859 | 1.2938 | 2.7704 | 3.3159 | 2.3329 | 1.246 | 3.06 | 2.83 | 1.68 | 1.71 | 1.25 | 0.99 | 2.6246E-07 | 3.6563E-06 | 4.6446E-06 | 5.6785E-06 | 2.9061E-06 | 1.2319E-06 |
| | | PFR | | | 2.836 | | 2.366 | | | | 0.21 | | 0.17 | | | | 5.946E-07 | | 3.893E-07 | |
| Rome | 1908 | IL | 0.0847 | 1.278 | 2.765 | 3.324 | 2.329 | 1.25 | 0.93 | 0.68 | 0.4 | 0.22 | 0.18 | 0.34 | 7.8752E-08 | 8.6904E-07 | 1.106E-06 | 7.3128E-07 | 4.1922E-07 | 4.25E-07 |
| | | XIL | 0.0862 | 1.298 | 2.7833 | 3.3328 | 2.3305 | 1.25 | 2.98 | 3.34 | 1.52 | 1.37 | 0.96 | 1.11 | 2.5662E-07 | 4.3291E-06 | 4.2278E-06 | 4.5499E-06 | 2.2329E-06 | 1.3820E-06 |
| | | PFR | | | 2.834 | | 2.369 | | | | 0.55 | | 0.25 | | | | 1.559E-06 | | 5.794E-07 | |



Atmospheric Measurement Techniques — Discussions — Open Access

| Location | ID | Method | 0.0841 | 1.26 | 2.747 | 3.315 | 2.32 | 1.246 | 2.34 | 1.63 | 0.96 | 0.62 | 0.47 | 0.43 | 1.9675E-07 | 2.0538E-06 | 2.637E-06 | 2.055E-06 | 1.0904E-06 | 5.3578E-07 |
|---|---|---|---|---|---|---|---|---|---|---|---|---|---|---|---|---|---|---|---|---|
| Rome | 1909 | IL | 0.0841 | 1.26 | 2.747 | 3.315 | 2.32 | 1.246 | 2.34 | 1.63 | 0.96 | 0.62 | 0.47 | 0.43 | 1.9675E-07 | 2.0538E-06 | 2.637E-06 | 2.055E-06 | 1.0904E-06 | 5.3578E-07 |
|  |  | XIL | 0.0866 | 1.309 | 2.823 | 3.3463 | 2.3356 | 1.2564 | 3.94 | 3.61 | 2.87 | 1.90 | 1.11 | 0.95 | 3.4160E-07 | 4.7260E-06 | 8.0981E-06 | 6.3714E-06 | 2.5963E-06 | 1.1914E-06 |
|  |  | PFR |  |  | 2.838 | 2.369 |  |  |  |  | 0.12 |  | 0.07 |  | 3.260E-07 |  | 1.754E-07 |  |  |  |
| Rome | 2108 | IL |  | 1.251 | 2.716 | 3.301 | 2.259 | 1.266 | 2.67 | 2.61 | 2.47 | 2.61 | 0.28 | 0.31 |  | 3.3402E-06 | 6.709E-06 | 8.616E-06 | 6.3252E-07 | 3.9246E-07 |
|  |  | XIL | 0.0854 | 1.2788 | 2.7291 | 3.2931 | 2.2788 | 1.2458 | 2.75 | 3.25 | 2.46 | 1.65 | 1.48 | 0.91 | 2.3456E-07 | 4.1561E-06 | 6.7215E-06 | 5.4234E-06 | 3.3797E-06 | 1.1382E-06 |
| Rome | 2109 | IL | 0.0818 | 1.232 | 2.686 | 3.268 | 2.25 | 1.25 | 1.23 | 1.35 | 0.61 | 0.52 | 0.54 | 0.39 | 1.006E-03 | 1.663E-02 | 1.638E-02 | 1.699E-02 | 1.215E-02 | 4.875E-03 |
|  |  | XIL | 0.083 | 1.251 | 2.7016 | 3.2964 | 2.2847 | 1.2526 | 2.17 | 2.73 | 2.29 | 2.87 | 2.99 | 2.50 | 1.8048E-07 | 3.4187E-06 | 6.1926E-06 | 9.4722E-06 | 6.8267E-06 | 3.1340E-06 |
|  |  | PFR |  |  | 2.754 | 2.302 |  |  |  |  | 0.22 |  | 0.44 |  |  | 6.179E-07 |  | 1.004E-06 |  |  |
|  |  | Cim_1270 | 0.085 |  |  |  |  |  |  |  |  |  |  |  | 1.250E-07 | 3.020E-06 | 3.020E-06 | 3.600E-06 | 2.510E-06 | 1.830E-06 |
| Davos | 2110 | IL | 0.0851 | 1.255 | 2.770 | 3.310 | 2.280 | 1.240 | 1.48 | 1.09 | 1.10 | 1.09 | 1.10 | 1.49 | 1.200E-03 | 6.401E-03 | 5.936E-03 | 3.598E-03 | 4.586E-03 | 7.436E-03 |
|  |  | XIL | 0.0862 | 1.2612 | 2.698 | 3.271 | 2.293 | 1.219 | 1.41 | 0.51 | 0.22 | 0.11 | 0.2 | 0.61 |  |  |  |  |  |  |
|  |  | PFR |  |  | 2.7043 | 3.2928 | 2.311 |  | 0.53 | 0.86 | 0.63 | 0.00 | 0.31 | 0.20 | 4.5824E-08 | 1.0882E-06 | 1.7002E-06 | 7.2490E-07 | 4.713E-07 | 3.628E-07 |
| PTB | 2206 | Lab | 0.0903 | 1.3225 | 2.9680 | 3.5506 | 2.4146 | 1.2473 | 4.4 | 4.3 | 4.2 | 4.2 | 4.1 | 4.2 | 4.000E-07 | 5.700E-06 | 1.300E-05 | 1.500E-05 | 1.000E-05 | 5.300E-06 |
| Izana | 2209 | SL | 0.0855 | 1.2551 | 2.6982 | 3.2715 | 2.2965 | 1.2372 | 2.53 | 1.09 | 0.40 | 0.15 | 0.45 | 0.66 | 2.160E-07 | 1.370E-06 | 1.090E-06 | 5.070E-07 | 1.040E-06 | 8.210E-07 |

Table 2a: Solar calibration constants $V_0$, percent Coefficients of variation CV, and uncertainties calculated as described from sections 3.1-3.6, for all the methods and periods, for POM_CNR. When CV or Unc is 0, the monthly dataset is composed by only one point. In column three, there is the type of method used: IL (Improved Langley), XIL (Cross Improved Langley), PFR (Transfer from PFR instrument), Cim_1270 (Transfer from Cimel), Lab (laboratory calibration), SL (Standard Langley).






| Site | YYMM | Method | V0_IL*e-04 (A) 340 | 400 | 500 | 675 | 870 | 1020 | %CV 340 | 400 | 500 | 675 | 870 | 1020 | Unc 340 | 400 | 500 | 675 | 870 | 1020 |
|---|---|---|---|---|---|---|---|---|---|---|---|---|---|---|---|---|---|---|---|---|
| Rome | 2109 | IL | 0.0118 | 0.7635 | 2.535 | 3.803 | 2.266 | 1.084 | 2.87 | 2.32 | 0.63 | 0.54 | 0.45 | 0.70 | 3.3859E-08 | 1.7683E-06 | 1.6029E-06 | 2.0540E-06 | 1.0129E-06 | 7.5869E-07 |
| Rome | 2109 | Cim_1270 | 0.0124 |  | 2.6149 | 3.8487 | 2.3072 | 1.0580 | 1.19 |  | 1.18 | 1.05 | 1.07 | 1.43 | 1.477E-08 |  | 3.082E-06 | 4.037E-06 | 2.471E-06 | 1.516E-06 |
| Rome | 2109 | PFR |  |  | 2.6153 |  | 2.3130 | 1.0889 |  |  | 1.38 |  | 1.12 |  |  |  | 3.6159E-06 |  | 2.5904E-06 |  |
| PTB | 2206 | Lab | 0.0123 | 0.7893 | 2.7770 | 3.9341 | 2.3583 | 1.0889 | 4.4 | 4.2 | 4.2 | 4.1 | 4.1 | 4.2 | 5.430E-08 | 3.280E-06 | 1.180E-05 | 1.610E-05 | 9.770E-06 | 4.520E-06 |
| Valen. | 2210 | IL | 0.0116 | 0.761 | 2.565 | 3.841 | 2.287 | 1.081 | 1.04 | 0.66 | 0.97 | 1.37 | 1.23 | 1.86 | 1.2027E-08 | 5.0584E-07 | 2.4780E-06 | 5.2583E-06 | 2.8130E-06 | 2.0128E-06 |
| Valen. | 2210 | XIL | 0.0117 | 0.7633 | 2.6103 | 3.8144 | 2.2878 | 1.0986 | 3.55 | 7.41 | 6.71 | 2.73 | 2.06 | 7.00 | 4.1569E-08 | 5.6586E-06 | 1.7509E-05 | 1.0429E-05 | 4.7225E-06 | 7.6943E-06 |
| Valen. | 2211 | IL | 0.0123 | 0.7804 |  | 3.873 | 2.32 | 1.081 | 1.61 | 2.05 |  | 0.54 | 0.66 | 1.67 | 1.9807E-08 | 1.5959E-06 |  | 2.0918E-06 | 1.5219E-06 | 1.8031E-06 |
| Valen. | 2211 | XIL | 0.0122 | 0.7841 | 2.6006 | 3.8652 | 2.3123 | 1.0574 | 1.22 | 0.00 | 0.17 | 0.38 | 0.48 | 0.00 | 1.4838E-08 | 0.00 | 4.4522E-07 | 1.4502E-06 | 1.1173E-06 | 0.00 |
| Valen/Izana | 2211 | SL_tranf | 0.0124 | 0.7776 | 2.5673 | 3.8002 | 2.3105 | 1.0753 | 2.58 | 1.11 | 0.44 | 0.26 | 0.50 | 0.73 | 3.2100E-08 | 8.6415E-07 | 1.1423E-06 | 9.9221E-07 | 1.1599E-06 | 7.8000E-07 |

Table 2b: Solar calibration constants $V_0$, percent Coefficients of variation CV, and uncertainties calculated as described from sections 3.1-3.6, for all the methods and periods, for POM_UV. When CV or Unc is 0, the monthly dataset is composed by only one point. In column three, there is the type of method used: IL (Improved Langley), XIL (Cross Improved Langley), PFR (Transfer from PFR instrument), Cim_1270 (Transfer from Cimel), Lab (laboratory calibration), SL_trans (Transfer from POM_CNR Standard Langley).



**3.3   The improved Langley methods (IL-XIL) for POM_CNR and POM_UV**
Based on the above-described Langley method, the formula of Improved Langley method is expressed as follows:

$$y = a_{IL} + b_{IL}x \quad \text{where}$$
$$x = m_0\omega\tau_{ext} = m_0\frac{\tau_{sca}}{\tau_{ext}}\tau_{ext} = m_0\tau_{sca} \quad and$$
$$y = \ln V + m_0 \cdot (\tau_{gas} + \tau_R) \hspace{4cm} \text{Eq.3}$$

where ω is the aerosol single scattering albedo (defined as $\frac{\tau_{sca}}{\tau_{ext}}$ ). The linear fitting provides intercept $a_{IL} = \ln V_0$ and slope
$b_{IL} = -\frac{1}{\omega}$.
The improved Langley plot method (Campanelli et al., 2004 and 2007, Nakajima et al., 2020) is the standard calibration
method of the SKYNET network and it was used to calculate the solar calibration constants for both the Prede-POM sun-
sky photometers.
The calibration value, $V_0$, is retrieved by fitting the natural logarithm of the direct solar irradiance versus the product of
$m_0$ and the scattering optical depth, as retrieved by the SKYRAD 4.2 code (Nakajima et al., 2020), instead of only the air
mass as occurs with the standard Langley plot.
To understand the main idea on which this method is based, we define the two observable quantities (for each wavelength
λ) important for the Sun-sky photometer, the direct solar irradiance in Eq. 1 and the normalized radiance R in Eq. 4

$$R(\Theta) = \frac{E(\Theta)}{\Delta\Omega \cdot V \cdot m_0} \hspace{4cm} \text{Eq. 4}$$

where $\Theta$ is the scattering angle at which the Prede-POM takes measurements of the sky diffuse irradiance $E$, V is direct
irradiance and $\Delta\Omega$ is the solid-view angle of the instrument.
R is determined as the solution of the radiative transfer equation, as in Eq.5 in the Almucantar geometry for a one-layer
plane-parallel atmosphere, where P is the phase function, and q indicates the multiple-scattering contribution

$$R(\Theta) = \omega\tau_{ext}P(\Theta) + q(\Theta) = \tau_{sca}P(\Theta) + q(\Theta) \hspace{3cm} \text{Eq. 5}$$

Thus, normalized radiance R is approximately assumed as the product of $\tau_{sca}$ and P; $\tau_{sca}$ is derived via the inversion process
(e.g., Skyrad 4.2) of volume size distribution from the normalized radiance in aureole region with scattering angles 3° <
$\Theta$ < 30° (Nakajima et al., 2020), keeping fixed the refractive index, and it is used in the improved Langley method for
obtaining the intercept $V_0$. Note that the aerosol optical depth for scattering (in $x$ in Eq. 3) is potentially retrieved more
accurately than the optical depth for extinction $\tau_{ext}$. To understand the reason, it must be considered that the volume size
distribution is roughly obtained by only direct radiation information because of the limited information content of the
extinction Kernel function (Tonna et al., 1995, Figure 4). On the other hand, for the sky radiance measurements in the
range 3° < $\Theta$ < 30°, the scattering kernel functions (Tonna et al., 1995, Figure 4) have reliable information content
(approximately within $1 < 2\pi r/\lambda < 60$, which means that $0.05 < r < 10$ μm for our wavelength set) that is sufficient for
deriving volume size distribution and reliably reconstructing the connected quantities $R, P, \omega\tau_{ext}$. The radiance in the
aureole region is also less sensitive to the refractive index (Tanaka et al., 1983), Therefore, the use of R in Eq.5 to obtain
$\omega\tau_{ext}$, i.e. scattering optical thickness, is the best way to analyze data. In contrast to the standard Langley method, the
intercept $V_0$ does not depend on the daily variability of $\omega\tau_{ext}$ if the inversion process is accurate.
From R and V data collected each month, two $V_0$ values a day are calculated with data taken in the morning and in the
afternoon, and the $V_0$ monthly means are quality checked according to Campanelli et al., 2007, and below summarized:
i) the values of $\omega\tau_{ext}$ obtained from the SKYRAD4.2 code inversion with accuracy lower than 7% are rejected. The
accuracy is estimated as the percent differences between the measured and retrieved radiance R, averaged over all the
wavelengths and scattering angles; ii) only the measurements taken for $m_0 < 3.0$ and $1/\omega > 0$ and $\leq 2$ are selected; iii) all
the values of $V_0$ found for $\tau_{ext}$ (500 nm) $\geq 0.4$ are rejected; iv) a minimum number of 10 points is used in each morning
and afternoon fit.
The rejection of $\tau_{ext}$ (500 nm) values greater than 0.4, is not in contradiction with the AERONET strategy, where the
retrieval of ω is performed only for $\tau_{ext} > 0.4$ (Aeronet web page, Holben et al., 2006) otherwise ω and other properties
are not included in the L2 analysis, because the purpose of this selection for IL is different. Infact a potential problem in
this procedure is that the refractive index is kept fixed. The aureole region has information for volume size distribution,
but not for refractive index, as said before, and this allows to retrieve $\tau_{sca}$. However, high $\tau_{ext}$ makes high multiple
scattering contribution ($q(\Theta)$ in Eq. 5) and greater error in retrieving $\tau_{sca}$ with a fixed refractive index.

Once the filtered monthly $V_0$ series are obtained, the outliers and short-term variations related to the method itself are
filtered using the Chauvenet criterion (H. D. Young, 1962), that rejects points out of 2 times the standard deviation (std),
and a three-point moving average technique. Finally, if at least 3 values remain and the ratio between their std and mean
(Coefficient of variation, CV) is <3%, the monthly mean $V_0$ value is calculated. The uncertainty related to this value is
given for each wavelength by the CV coefficient. Results are shown in Table 2.
In the real observations, it is difficult to separate natural variations and inversion errors of $\omega\tau_{ext}$ and thus undesired
inversion errors can be included that lead the IL method to an underestimation of the fitting parameters in the case of
large aerosol retrieval errors (Nakajima et al., 2020). A new solution to this problem is tested, named the cross IL method
(XIL), which exchanges the role of x and y in the regression analysis as described in Eq. 6

$$x = a_{XIL} + b_{XIL}y \qquad\qquad\qquad \text{Eq. 6}$$

The linear fitting provides slope $b_{XIL} = \frac{1}{b_{IL}} = -\omega$ and intercept $a_{XIL} = -\frac{a_{IL}}{b_{IL}} = \omega ln\, V_0$
The selection of data for this method is performed using the threshold of 0.05 for the fitting error, assuming that
retrieval errors on $\omega$ and $\tau$ from Skyrad are within 9% (Nakajima et al., 2020). Monthly $V_0$ and the corresponding %CV
are then calculated. Results are shown in Table 2.
**3.4 The standard Langley method Transfer from POM_CNR to POM_UV**
The calibration of the Prede POM_CNR by the Standard Langley Plot method at Izaña campaign in September 2022, was
transferred to POM_UV using data from the QUATRAM3 campaign, on September 2021, as it was the only campaign
where both instruments were co-located.
After visual inspection of the signal ratios for the days of September 2021, the days in the intervals 4-9, 11-15, 17-19, are
considered for the calibration transfer.
The transfer procedure consisted of the following steps: i) data were selected between 9 to 13 UTC; ii) signals within 30
sec between POM_UV and POM_CNR were considered; iii) $V_0$ for POM_UV was calculated following Eq. 7:

$$V_0^{POM\_UV} = V_0^{POM\_CNR} \cdot \frac{V^{POM\_UV}}{V^{POM\_CNR}} \qquad\qquad \text{Eq.7}$$

iv) values that are more than three scaled median absolute deviations away from the median are assumed as outliers and
deleted; v) daily $V_0^{POM\_UV}$ medians are calculated and 2std of the $V_0^{POM\_UV}$ series is calculated. If 2std is larger than 0.5%
of the daily $V_0$ median, all data outside 2std are removed. The process is repeated until 2std becomes equal or smaller
than 0.5% of the daily $V_0^{POM\_UV}$ median or standard deviation and median values becomes equal in continuous iteration;
vi) after visual inspection only days were selected which are stable, resulting in the exclusion of the days stated before.
To calculate the uncertainty of the transferred calibration values, the equation below was used, where we account for
uncertainties on the master instrument calibration, and the standard deviation of the signal ratios, that are sensitive to
changes in AOD, etc.

$$u(V_0^{POM\_UV}) = V_0^{POM\_UV} \cdot \sqrt{\left(\frac{u(V_0^{POM\_CNR})}{V_0^{POM\_CNR}}\right)^2 + \left(\frac{STD(SR)}{SR}\right)^2} \qquad \text{Eq.8}$$

where: $V_0^{POM\_UV}$ is the mean of the calibration values series and $u(V_0^{POM\_UV})$ is the uncertainty associated; $V_0^{POM\_CNR}$ is
the calibration factor, and $u(V_0^{POM\_CNR})$ is the uncertainty associated with it. This uncertainty was estimated as the
standard deviation of the 6 calibration values obtained by the 6 plots used in section 3.2; $SR$ is the ratio of signals
$\left(\frac{V^{POM\_UV}}{V^{POM\_CNR}}\right)$ and $STD(SR)$ is the standard deviation of the ratio of the signals available for the calibration. Results are in
Table 2.
**3.5 The calibration transfer from PFR to POM_CNR and POM_UV**
The transfer of calibration from two reference PFR photometers of the PMOD one located in Davos and the other in
Rome, has been carried out for both POM_CNR and POM_UV, during the QUATRAM campaigns.
The transfer is based on the ratio of Eq. 9 for the two instruments, POM and PFR:

$$\frac{V_0^{POM,TR}}{V_0^{PFR}} = \frac{V^{PFR}}{V^{POM}} \qquad\qquad \text{Eq.9}$$

where, $V^{PFR}$ and $V^{POM}$ are the solar direct irradiance measured by the two instruments, $V_0^{POM,TR}$ is the unknown solar
calibration constant of the POM and $V_0^{PFR}$ the known calibration constant of the PFR to be transferred. For QUATRAM
3 in Rome, days in the intervals (6-8; 11-14) of September 2021 were considered.
Signals ratios $\frac{V^{PFR}}{V^{POM}}$ were taken using measurements that are within 30 sec time difference, and cloudy conditions were
removed, together with ratios outliers. Values out of the interval time 9-13 UTC were rejected. The time interval was
chosen as 9-13 to avoid the rapid change in airmass. From Eq.9 the time series of $V_0^{POM,TR}$ was limited to:  i) choosing
only those days for which at least 20 measurements in 1 hour are available; ii) Calculating the daily $V_0$ medians and
compare each with 2std of the day's $V_0$ values; if 2std is larger than 0.5% of the daily $V_0$ median, remove all data outside
2std; repeat until 2std becomes equal or smaller than 0.5% of the daily $V_0$ median; when this is accomplished, if the day's
measurements have dropped below 20 the day is excluded. Daily medians of the remaining values are calculated, and
then a monthly mean $V_0^{POM,TR}$  is estimated. As uncertainty the std of the monthly mean values is assumed. Results are in
Table 2.
For the transfer to POM_UV during QUATRAM 3, the same procedure was applied but the selected days are in intervals
(6-9; 11-14) of September 2021.
The uncertainties were estimated as in other transfer cases, by assuming a nominal uncertainty of the PFR calibration of
1%. Results for both instruments are in Table 2.
The same procedure was applied for the QUATRAM 3 in Davos and QUATRAM 1 and 2 in both the sites for POM_CNR.
### 3.6  Calibration transfer from CIMEL to POM_CNR and POM_UV

During QUATRAM 3, a calibration transfer from the Cimel #1270 was carried on, following the same selection criteria
of the transfer from PFR.
To calculate the total uncertainty of the transferred calibration values, Eq. 8 was used with $V_0^{CIM}$ as the master instrument
and $u(V_0^{CIM})$ the associated uncertainty. As the estimated uncertainty is absent for the master instrument, it is assumed to
nominal 1% of $V_0$. Results are in Table 2.
### 3.7  Comparisons

*a)  Differences between all methods against the reference one*
The six calibration methods described in the above sections in the period September 2021- November 2022 for both the
POMs are compared against a reference calibration. The time interval was chosen because the campaigns and laboratory
calibrations were performed in this period in the framework of the MAPP project.
For the POM_CNR the reference calibration is the Standard Langley method performed at Izana in September 2022,
whereas the transfer of this calibration to the POM_UV is the reference value for the latter instrument. However, we need
to consider that the frequent shipments of the equipment during this year for the project purpose and the usage can have
affected the values of $V_0$ and probably can be the reason of discrepancies between the SL calibration and the calibrations
performed about 1 year earlier. The aging of the instrument, without shipments, can also affect the $V_0$ but the order of
magnitude and amount per year strongly depends on the instrument, and some wavelengths can be more affected than
others. For the two instruments used in this work it is not possible to evaluate a degradation in one year and discern it
from the shipment's effects, because the equipment was frequently travelling.
The percent difference was calculated with Eq.10:
$$Diff(\%) = \frac{(V_0^{ref} - V_0^x)}{V_0^{ref}} \cdot 100 \qquad\qquad Eq.\ 10$$
where $V_0^{ref}$ is the reference value and $V_0^x$ is the calibration obtained with each of the above-described methods. Results
are shown in Figure 4 and Table 3.
For the POM_CNR the agreement is very good with the reference SL and many of the points are within ±1%.
The agreement generally improves with the wavelengths but with a small worsening at 1020 nm. The transfer from Cimel
and PFR in Rome and from PFR in Davos at 500 nm differ of -1.6%, -2.1% and -1.3%, respectively. 340 nm is the
wavelength with the most problematic results for the on-site procedures in Rome (differences around 4%). Further studies,
not yet published, showed that the 340 nm is also significantly affected by the assumed surface Albedo, and improvements
of the agreement were found if, for example, values from the POLarization and Directionality of the Earth's Reflectances
radiometer (POLDER) on ADEOS satellite, are considered.  More tests are neede to verify this dependence in for more
sites. Moreover, according to Momoi (2022) the molecular polarization potentially causes calibration errors from IL and
XIL methods at the UV region (340 nm), especially low aerosol loading atmosphere. In fact the SKYRAD.pack 4.2 used
for the on-site procedures has an un-polarized ("scalar") radiative transfer core forward model, that can cause around 8%
errors on the retrieval of radiance at 340 nm, so it might be one of the reasons for the calibration constant of 340 nm to
have errors.
The best agreement is for the IL in Davos with values < 0.5% at all the wl, and 1.5% at 1020 nm.
For the POM_UV, many points are within ±1% but less respect to the POM_CNR.  The agreement with the reference
method for the PTB laboratory calibration shows an improvement, remaining however between -1.3% and -8% except





for the 340 nm where it is 0.7%. Also for the POM_UV, an improvement with the wavelengths is notable with a worsening
at 1020 nm. The transfer from Cimel and PFR in Rome at 500 nm agrees within -1.9 %, a value comparable with those
of the POM_CNR. Also, in this case the 340 nm is the wavelength with the most problematic results for the on-site
procedures (differences up to 6%) as explained for the POM_CNR.
For both POMs, the comparison with PTB calibration shows very high underestimations (down to -10% except for
POM_CNR, and -8% for POM_UV), but at this state of the art we are not able to provide a certain reason for the
discrepancy. It is noteworthy that the agreement between the laboratory calibration and the Langley measurements for
PFR was well consistent within the uncertainties. In the case of the CIMELs, however, discrepancies increasing towards
the short wavelengths and exceeding the uncertainties by a factor of 2-3 have been observed. The causes of the
discrepancies between the laboratory calibrations and the field measurements of the CIMEL and POM instruments are
not yet understood. The instruments are obviously aligned and operated using different procedures when calibrated in the
laboratory and when measuring in the field. CIMEL and especially POM have narrower field of views than the PFR,
which makes them more susceptible to alignment and tracking errors, which could possibly lead to systematic
underestimation of the measured irradiance values. It should be noted that the comparison results shown in Figure 4 are
all from relative (Langley) measurements, with the exception of those based on the absolute responsivity calibrations at
PTB, which makes the respective result in the comparison particularly sensitive to the effects mentioned above.
Focusing on the on-site methodologies, the IL works better in Davos with an agreement against SL always below 0.5%
except at 1020 nm where it increases up to about 1.5%. A very good accordance is also found in Valencia in November
2022, always within 0.8% except at 500 and 675 nm (within 1.5%). The similarity between the two cases is probably due
to the very low turbidity recorded in this month in Valencia, that makes the atmosphere optically more similar to the one
in Davos.
The XIL provides a consistent improvement, with values within 1%, only in Rome for all the wavelengths, but in very
clean atmosphere, as in Davos, it was not possible to retrieve values at 1020 nm, as conversely is done with the IL. This
is related to the differences in the data screening criteria between the two methods, set up for performing the linear fitting.

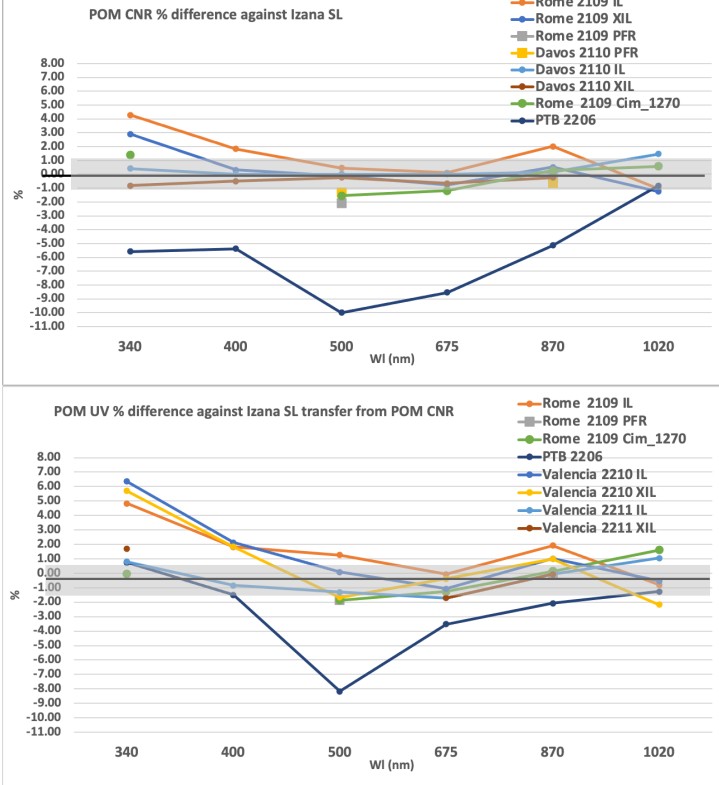


Figure 4. The percent coefficients of variation, calculated as the % ratio between the standard deviation and the mean
values.


Table 3.  % Differences between five calibration methods and the reference one and % CV of the retrieved V₀.

| | | POM_CNR | % difference | | | | | | % CV | | | | | |
|---|---|---|---|---|---|---|---|---|---|---|---|---|---|---|
| | | | 340 | 400 | 500 | 675 | 870 | 1020 | 340 | 400 | 500 | 675 | 870 | 1020 |
| Rome | 2109 | IL | 4.29 | 1.84 | 0.45 | 0.11 | 2.02 | -1.04 | 1.23 | 1.35 | 0.61 | 0.52 | 0.54 | 0.39 |
| Rome | 2109 | XIL | 2.91 | 0.32 | -0.13 | -0.76 | 0.51 | -1.25 | 2.17 | 2.73 | 2.29 | 2.87 | 2.99 | 2.17 |
| Rome | 2109 | PFR | | | -2.08 | | -0.25 | | | | 0.22 | | 0.44 | |
| Rome | 2109 | Cim_1270 | 1.39 | | -1.55 | -1.18 | 0.28 | 0.58 | 1.48 | | 1.10 | 1.09 | 1.10 | 1.49 |
| Davos | 2110 | PFR | | | -1.33 | | -0.63 | | | | 0.13 | | 0.20 | |
| Davos | 2110 | IL | 0.42 | 0.00 | 0.01 | 0.02 | 0.15 | 1.47 | 1.41 | 0.51 | 0.22 | 0.11 | 0.20 | 0.61 |
| Davos | 2110 | XIL | -0.83 | -0.49 | -0.23 | -0.65 | -0.24 | | 0.53 | 0.86 | 0.63 | 0.00 | 0.31 | |
| PTB | 2206 | | -5.58 | -5.37 | -10.00 | -8.53 | -5.14 | -0.82 | 4.4 | 4.3 | 4.2 | 4.2 | 4.1 | 4.2 |
| | | POM_UV | 340 | 400 | 500 | 675 | 870 | 1020 | 340 | 400 | 500 | 675 | 870 | 1020 |
| Rome | 2109 | IL | 4.82 | 1.81 | 1.26 | -0.07 | 1.93 | -0.81 | 2.87 | 2.32 | 0.63 | 0.54 | 0.45 | 0.70 |
| Rome | 2109 | Cim_1270 | -0.04 | | -1.86 | -1.27 | 0.14 | 1.61 | 1.19 | | 1.18 | 1.05 | 1.07 | 1.43 |
| Rome | 2109 | PFR | | | -1.87 | | -0.11 | | | | | | | |
| PTB | | | 0.74 | -1.50 | -8.17 | -3.52 | -2.07 | -1.26 | 4.4 | 4.2 | 4.2 | 4.1 | 4.1 | 4.2 |
| Valencia | 2210 | IL | 6.35 | 2.14 | 0.09 | -1.07 | 1.02 | -0.53 | 1.04 | 0.66 | 0.97 | 1.37 | 1.23 | 1.86 |
| Valencia | 2210 | XIL | 5.71 | 1.84 | -1.68 | -0.37 | 0.98 | -2.16 | 3.55 | 7.41 | 6.71 | 2.73 | 2.06 | 7.00 |
| Valencia | 2211 | IL | 0.79 | -0.36 | | -1.91 | -0.41 | -1.16 | 2.05 | | 0.54 | 0.66 | 1.67 | 2.05 |
| Valencia | 2211 | XIL | 1.68 | -0.84 | -1.30 | -1.71 | -0.08 | 1.05 | 1.22 | | 0.17 | 0.38 | 0.48 | |

475        *a) Long term differences between on-site calibrations and PFR transfer*

The difference between the on-site calibration methods and the PFR calibration transfer was analyzed in the period of the
3 QUATRAM campaigns held in Davos and Rome using Eq.10 with $V_0^{ref}$ the transfer from PFR. V₀s are shown in Table
2 and the percent difference is in Table 4 and Figure 5.
Table 4. % Differences between PFR transfer of calibration and the on-site calibration methods at the common
wavelengths.

| | | | % diff 500 nm | | % diff 870 nm | |
|---|---|---|---|---|---|---|
| | | Date | IL | XIL | IL | XIL |
| | | **1708** | 0.56 | 0.20 | 0.09 | -0.12 |
| | | **1807** | 1.08 | 0.19 | 1.08 | 0.20 |
| | **DAVOS** | **1808** | 1.32 | 0.39 | 1.19 | 0.89 |
| | | **1809** | 1.28 | 0.62 | 1.06 | -0.27 |
| | | **1810** | 1.28 | 0.82 | 0.76 | 0.33 |
| | | **2110** | 1.32 | 1.09 | 0.78 | 0.39 |
| | | **1710** | 2.66 | 2.72 | 0.99 | 0.40 |
| | | **1905** | 3.10 | 2.05 | 1.15 | 2.65 |
| | | **1906** | 2.81 | 1.17 | 1.87 | 0.60 |
| | **ROME** | **1907** | 3.49 | 2.31 | 2.83 | 1.40 |
| | | **1908** | 2.43 | 1.79 | 1.69 | 1.63 |
| | | **1909** | 3.21 | 0.53 | 2.07 | 1.41 |
| | | **2109** | 2.47 | 1.90 | 2.26 | 0.75 |






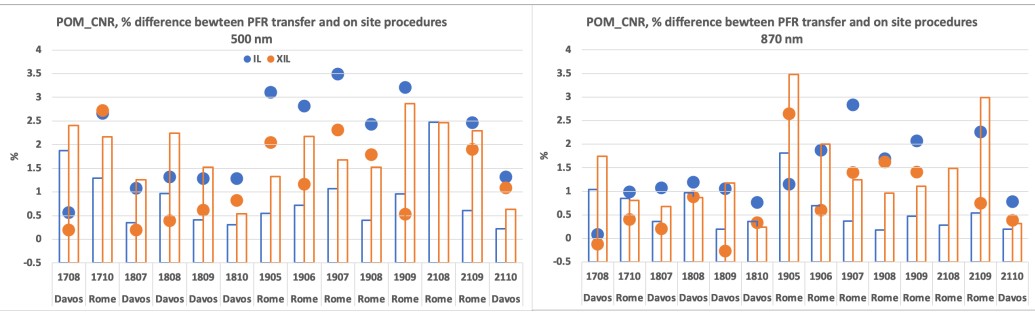

Figure 5. Percent differences between PFR transfer of calibration and the on-site calibration methods at the common
wavelengths (circles), and the uncertainty %CV of the IL and XIL as in Table 2.
For the IL, the differences are always greater than the uncertainties (%CV) of the method, for both wavelengths, with the
exception of Davos in 2017. Values are around 1% in Davos and this is an important result for the validation of the IL
procedure, confirming the good performance of the Improved Langley on high mountain even if, as shown in Nakajima
et al. (2020), the IL accuracy is proportional to the optical thickness of the atmosphere of observation, generally low on
high mountains. The same result has been also obtained by Ningombam et al. (2014). The largest differences are in Rome
and at 500 nm, although the higher AOD as shown in Figure 6. The reason could be related to the fact that in the retrieval
of $x$ for performing the fit in Eq.3, $\omega\tau_{ext}=\tau_{sca}$ and the refractive index must be assumed to not largely change during the
Langley plot (Campanelli et al., 2004), otherwise the retrieved optical thickness can include an error caused by the
inversion process and also by an improper pre-assigning of the refractive index. In an urban site, as Rome, we can expect
this assumption not satisfied. Further studies are actually aimed to understand the possibility of defining some selection
criteria for the variability of $\tau_{sca}$ values particularly in urban sites. Moreover the use of the Skyrad_MRI (Kudo et
al.,2021) instead of Skyrad 4.2 and possibility to use only the XIL method instead of the IL, is underevaluation.
For XIL many differences are within the uncertainties (%CV) of the method, and those higher are closer to the %CV
values than in the IL method. XIL improves the agreement particularly in Rome where the largest difference reduces from
3.5% to 2.5% at 500 nm and from 3% to 1.7% at 870 nm.

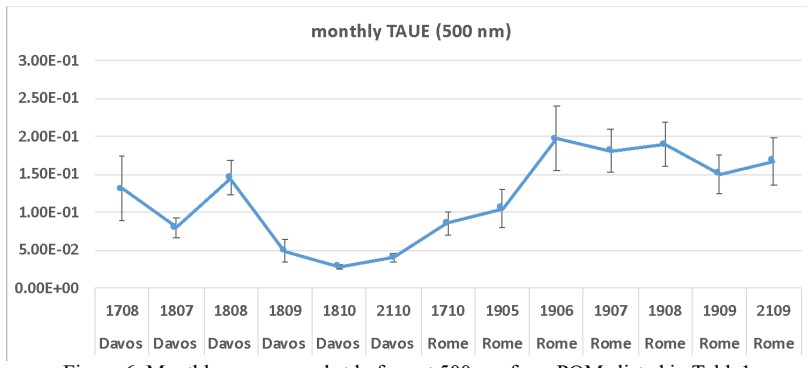

Figure 6. Monthly average and std of $\tau_{ext}$ at 500 nm from POMs listed in Table1.

**4.  Estimation of the solid view angle (SVA)**
The SVA is the measure of the field of view of the instrument that can be assumed from the geometry of the telescope.
However, several factors contribute to this value: color aberration of the lens, diffraction at the edges, misalignment of
the optical axis, and surface nonuniformity of filters and sensor. This makes it necessary to develop laboratory and on-
site methods for correctly estimating SVA values.  The methods used in this work are described below.

**4.1  Calibration at the laboratory of the AALTO University**
The field of view of the Prede POM_CNR has been measured at the laboratory of  Aalto University. The measurement
setup consists of a two-axis gimbal and a light source. The light source is constructed from an integrating sphere
(Gigahertz Optik type UMBB-300) and a 1 kW Xe-lamp. The diameter of the sphere is 300 mm, and the output aperture
is limited to 10 mm in diameter. The distance $D$ between the sphere aperture and the axis of rotation was $\approx 1060$ mm
(Figure 7). The purpose of the integrating sphere is to obtain a spatially uniform, well-defined light source. The aperture
size and the distance $D$ chosen provide the radiometer to see the light source at a solid angle corresponding to the same
solid angle where it sees the Sun in the field measurements, angular diameter = 0.54˚.

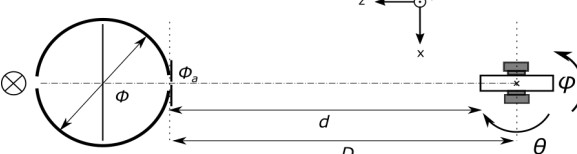


Figure 7: Schematic of the measurement setup. From left to right: a switchable light source, an integrating
sphere, and a two-axis gimbal.

The radiometer is mounted on the gimbal, tilted in the desired angle, and the signal amplitude is measured. The setup is
built on an optical rail, which enables easy varying of the distance between the gimbal and the light source. The light
source and gimbal are fixed in place. The point of rotation of the radiometer was chosen using an $x$-axis translator, and
customized elevation blocks installed between the radiometer and the gimbal to set $y$-direction. The common optical axis
of the light source and the radiometer is found by shifting the sphere aperture. The tilt angle range of measurements is [-
0.7˚ 0.8˚] for all channels in both directions, and the step size is 0.1°. The measurement sequence and the data collection
are automated using LabView. The integrating sphere and the Xe-lamp are shown in Figure 8.

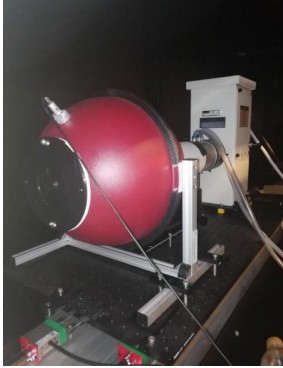

Figure 8. The integrating sphere with an interchangeable aperture and a monitor detector attached. The Xe-lamp housing
can be seen behind the sphere. Between the light source and the sphere there is a water filter and a lens imaging the arc
to the entrance of the sphere.

Collected data are used to derive the SVA of the POM following the method Boi et al. (1999). The solid viewing angle,
from the scanning centered at the origin of a local system of rectangular coordinates, is given by Eq. 11

$$\Delta\Omega = \iint_{\Delta A} \frac{E(x, y)}{E(0, 0)}\,\mathrm{d}x\mathrm{d}y \qquad \text{Eq. 11}$$

where E is the measured intensity (mA) and x and y (in radians) are the polar coordinates that determine the position of
the optical axis with respect to the position of the light source. The signals are registered as a function of the (x, y)
coordinates and a circular symmetry for the angular responsivity is assumed. Then a new system of coordinates centered
at the center of mass of the angular response is introduced and the needed parameters are obtained by fitting the
measurements.

The results are presented in Table 5, and in Figure 9 example of measurements are shown. The left figures display a 2D
heat map of the relative signal amplitude as a function of the two tilt angles. The fluctuations of the light source have
been taken into account by using correction coefficients obtained from the monitor detector data. The right figures present
the signal intensity as a 1D function of distance (r) from the center of mass. Measurements are particularly noisy, and it
is probably due to the use of an integrating sphere as source of light for a photometer, providing low radiation levels to
which the instrument has low sensitivity.

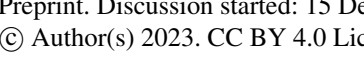



Table 5. SVA values and their uncertainties, obtained by laboratory calibrations and solar disk scanning methods

| | | SVA *e^{-04} (1/sr) | | | | | | | Unc*e^{-04} (1/sr) | | | | | | |
|---|---|---|---|---|---|---|---|---|---|---|---|---|---|---|---|
| | | 340 | 400 | 500 | 675 | 870 | 940 | 1020 | 340 | 400 | 500 | 675 | 870 | 940 | 1020 |
| POM_CNR AALTO | | 2.666 | 2.464 | 2.424 | 2.430 | 2.418 | 2.532 | 2.503 | / | / | / | / | / | / | / |
| POM_UV PMOD | | 2.198 | 2.298 | 2.302 | 2.343 | 2.396 | 2.433 | 2.382 | 0.016 | 0.011 | 0.009 | 0.012 | 0.012 | 0.009 | 0.011 |
| POM_CNR ROME | 3m | 2.4223 | 2.4633 | 2.4713 | 2.4588 | 2.5018 | 2.5038 | 2.5128 | 0.0144 | 0.0171 | 0.0190 | 0.0070 | 0.0056 | 0.0072 | 0.0090 |
| | 3n | 2.4363 | 2.4770 | 2.4825 | 2.4713 | 2.5255 | 2.5383 | 2.5425 | 0.0139 | 0.0171 | 0.0182 | 0.0071 | 0.0042 | 0.0063 | 0.0075 |
| POM_CNR IZANA | 3m | 2.3750 | 2.4370 | 2.4470 | 2.4382 | 2.4682 | 2.4882 | 2.4973 | 0.0680 | 0.0119 | 0.0084 | 0.0109 | 0.0507 | 0.0193 | 0.0196 |
| | 3n | 2.3813 | 2.4452 | 2.4538 | 2.4482 | 2.4798 | 2.5183 | 2.5258 | 0.0677 | 0.0122 | 0.0085 | 0.0124 | 0.0565 | 0.0196 | 0.0210 |
| POM_UV VALENCIA | 3m | 2.2528 | 2.3110 | 2.3368 | 2.3598 | 2.3923 | 2.4530 | 2.3910 | 0.0107 | 0.0143 | 0.0224 | 0.0222 | 0.0293 | 0.0197 | 0.0199 |
| | 3n | 2.2645 | 2.3180 | 2.3468 | 2.3708 | 2.4463 | 2.5040 | 2.4220 | 0.0090 | 0.0154 | 0.0222 | 0.0235 | 0.0309 | 0.0170 | 0.0217 |
| POM_UV ROME | 3m | 2.3080 | 2.3585 | 2.3625 | 2.3885 | 2.4770 | 2.5460 | 2.4720 | 0.0368 | 0.0092 | 0.0361 | 0.0396 | 0.0120 | 0.0410 | 0.0269 |
| | 3n | 2.2910 | 2.3475 | 2.3505 | 2.3770 | 2.4215 | 2.4940 | 2.4410 | 0.0438 | 0.0120 | 0.0389 | 0.0417 | 0.0170 | 0.0311 | 0.0240 |


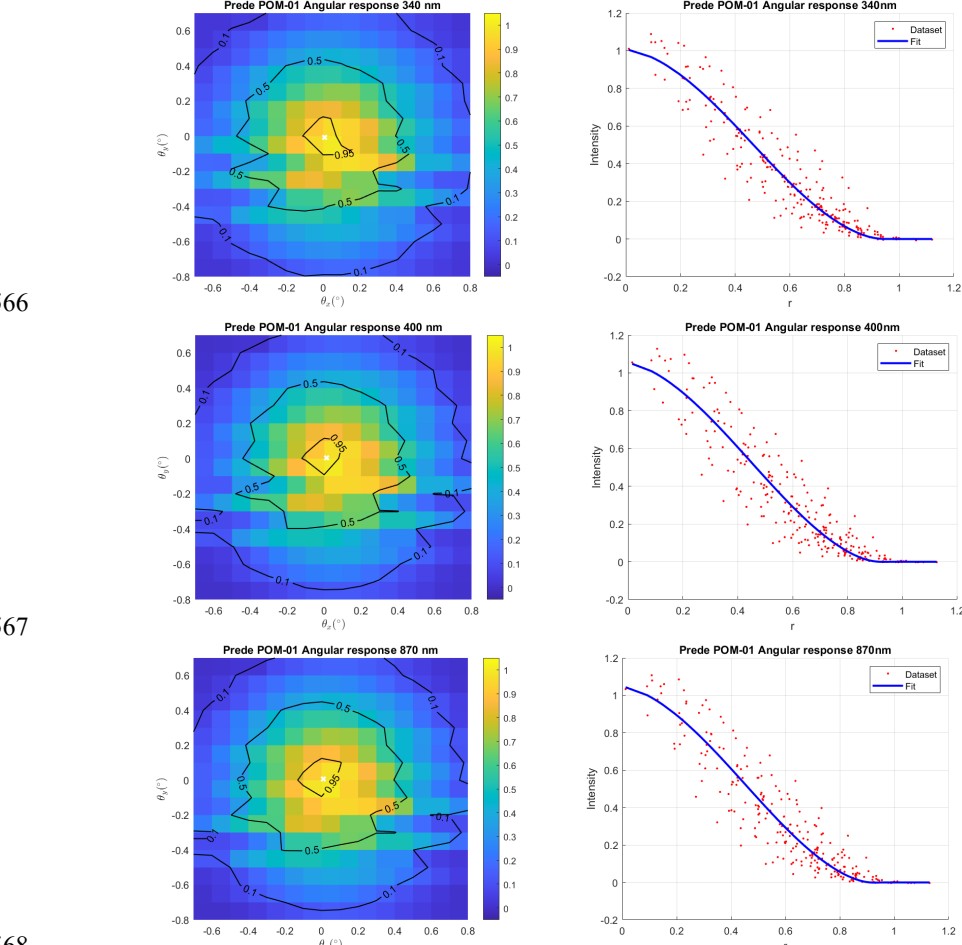

Figure 9. Normalized angular responsivities for the POM_CNR. Heatmaps on the left have been normalised to the
maximum intensity. Graphs on the right have been normalised to the average intensity within r<0.19° where the
responsivities were assumed to forma a plateau.





### 4.2 Calibration at the laboratory of PMOD

The field of view characterisation facility at PMOD/WRC consists of a 250-kW Xe-Lamp source and a 2-axis goniometer system with 0.2-mdeg resolution. The radiation from the Xe-Lamp shines on a Spectralon reflectance plate which produces a lambertian radiation distribution. An aperture with diameter 12 mm is placed in front of the reflectance plate, which is at a distance of 3600 mm from the goniometer system. Thus, the source has an apparent diameter of 0.19°. The field of view measurement consists rotating the radiometer head in both axes from -1.1° to + 1.1° in steps of 0.04°. At each position, the average of 10 measurements is stored, and every 100 positions, a reference measurement at the nominal center position (0°, 0°) is performed to monitor the stability of the source and of the radiometer. A whole measurement cycle for one channel of the radiometer takes 4.5 hours. The field of view (fov) of the instrument is obtained by normalising the measurements at every angle with the reference signal at (0°,0°), obtained by interpolating the reference measurements to the times of the individual measurements. For the measurements of this radiometer, the variability of the reference measurements varied by 0.38% during the whole measurement cycle. The field of view measurement of the Prede-POM_UV for the 500 nm channel is shown in Figure 10. As can be seen in the figure, the region with highest responsivity above 99% of the maximum is circular, with a diameter of approximately 0.5°.

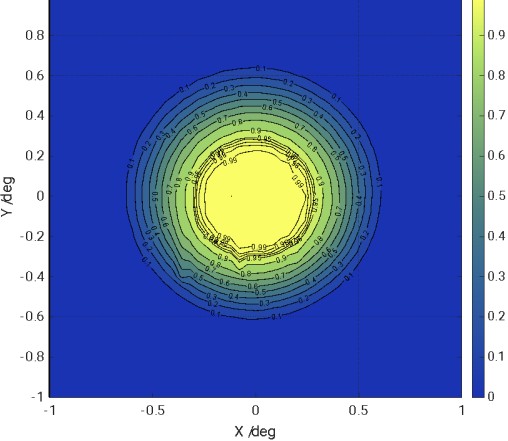

Figure 10: Field of view measurement at the 500 nm spectral channel of the Prede POM_UV. The measurements were normalised to the maximum signal.

From these measurements, the solid view angle $\Omega$ of the radiometer at this spectral channel is obtained by Eq. 11. The standard uncertainty of the solid angle measurements is obtained from the variability of the individual measurements, combined with the variability of the system obtained from the monitoring signals as described above. For the Prede POM_UV, the standard relative uncertainty of the solid angle determined from these measurements is 0.5%. Table 5 summarises the solid angle measurements determined for all spectral channels.

### 4.3 The solar disk methods

A methodology based on the scanning of the Solar disk, described in Boi et al. (1999) is used to determine SVA directly from optical data. It consists of the scanning of the irradiance field around the Sun, centered at the origin of a local system of a rectangular domain 2° by 2°; the irradiance is measured for all the channels at 21 x 21 gridded points around the solar disk with an angular resolution 0.1° (Figure 11 a, b). The instrument automatically follows the sun during the scanning, lasting several minutes, and measurements are corrected for the movement of the solar disk. The solid viewing angle, from the scanning centered at the origin of a local system of rectangular coordinates, is given by Eq 11. An elliptical system of coordinates centered at (0,0) is introduced and the needed parameters are obtained by fitting the measurements. This method is called solid3m hereafter.



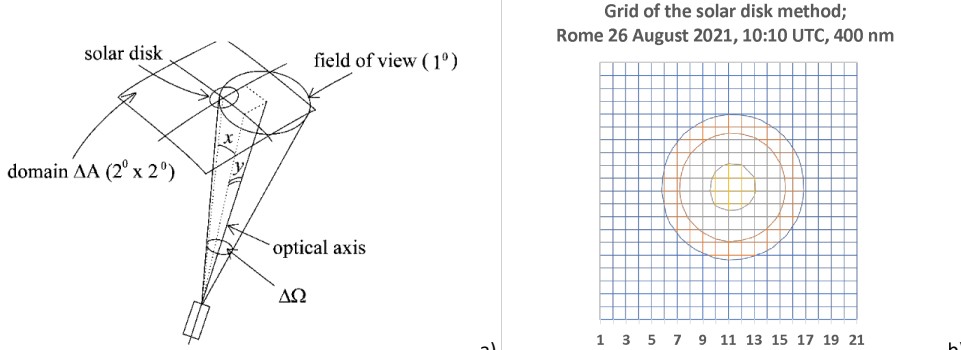

Figure 11. Geometry of the solar disk scanning measurements (a) and 2D image of the scanning.

The field of view of a PREDE-POM is 1°, the size of the sun disk is about 0.5°, and the rectangular domain is 2°x2°,
therefore the data are taken from the sun for scattering angles up to 1.4° (= (1°) ×$\sqrt{2}$). As shown in Uchiyama et al.,
(2018) the influence of the direct solar irradiance as a light source extends up to 2.5°. To take this into consideration,
the integration of Eq 11 is performed by linear extrapolation for angles larger than 1.4°. Before starting the data
processing, the minimum measured value is subtracted from the measured values, then the values between 1.4 and 2.5°
are extrapolated.
The above-described method has been implemented by Uchiyama et al., (2018), (hereafter called solid3n) by not
subtracting the minimum value largely affecting the measurements of the scattering angle between 1 and 1.4° and
extrapolating the values between 1.4° and 2.5° using the data from 1.0° to 1.4°.
SVA was calculated with the two solid3m and solid3n methods, using measurements taken in Rome and Valencia for
the POM_UV and in Rome and Izana for the POM_CNR. The errors (ERR) for both 3m and 3n methods are estimated
as $((AM/ZM)-1)^2$ where AM is the measure and ZM is the calculated signals during the fitting phase. Only SVA having
ERR <0.2 is selected. The mean value over each campaign is assumed as the final SVA, and its std as the uncertainty
associated to the estimation. Results are in Table 5.
The behavior of SVA values along the time, for the two methods (dashed lines is 3m and solid lines is 3n) and the two
instruments, was also analyzed in order to evaluate the stability of the method (Figure 12). The coefficient of variation
for the temporal variation (Std/mean) ranges from 1.1 to 1.3% for the POM_UV and from 0.7 to 0.9% for the POM_CNR
with the exception of 340 nm (2.5%) and 870 (2.0%) due to the point of September 3 out of the general patter for 340
and 870 nm.

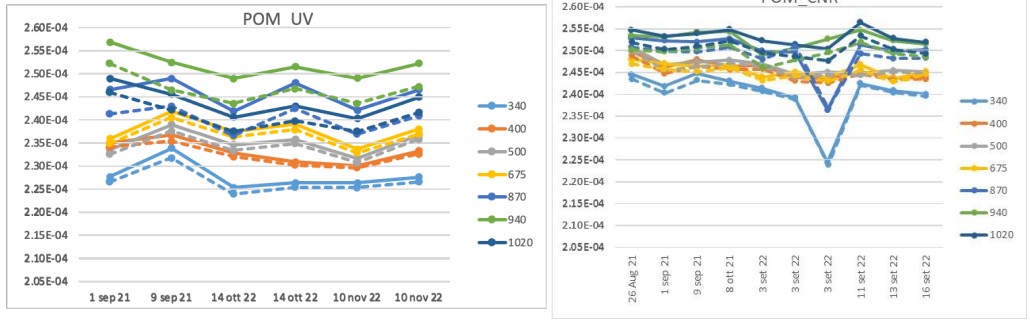

Figure 12: Temporal behaviour of SVA values [sr] from solid3m and 3n methods for POM_UV and POM_CNR co-
located in Rome.

**4.4 Comparisons**
SVA calculated with the two solid3m and solid3n methods, using measurements taken in Rome, Valencia, Davos and
Izana, are compared for both POM_UV and POM_CNR instruments against the laboratory calibrations performed in
AALTO and PMOD (Table 6 and Figure 13).





$$Diff(\%) = \frac{(SVA - SVA^{lab})}{SVA^{lab}} \cdot 100 \qquad \text{Eq.12}$$

The solar disk scanning method uses the sun direct irradiance measurements as light source whereas the radiance from an
integrating sphere is the source at Aalto laboratory providing lower radiation levels and noisy measurements as already
mentioned in the paragraph 4.1. This is probably the reason why for the 340 nm channel of the POM_CNR, the
wavelength with the lowest intensity level, a large discrepancy is found ranging from 8.62% to 10.92% in Rome and
Izana.
The solar disk scanning in Rome and Izana analyzed with the solid3m method agrees generally better, with respect to
solid3n, with the laboratory calibration. The difference varies from a minimum of 0.03% at 400 nm to a maximum of
3.46% at 870 nm in Rome and from 0.23% at 1020 nm to 2.07% at 870 nm in Izana. Both the methods slightly
overestimate the SVA values in Rome. The 870 nm shows the highest discrepancy in both the sites and for both the
solid3m and 3n methods. At this moment we are not able to provide a reason for it, even if we expect it is due not to a
physical cause, but to an instrumental one. A general overestimation by the onsite procedures in the range [500-870] nm
wavelengths is observed in both the sites. The overestimation is explained considering that the field of view of a PREDE-
POM is 1° and the size of the sun disk is about 0.5°, therefore the scattered light from aerosols and air molecules is
included in the measurement of the direct solar irradiance. Moreover, the direct solar light strikes the lens and results in
"stray" light. The scattering contribution and stray light reaching the detector increase the output, and the integrated value
has a larger magnitude that can affect the estimation of the SVA. The overestimation is lower in Izana due to a less
important scattering effect.
For the POM_UV, as for the other one, the 340 nm wavelength has a larger disagreement respect to the other wavelengths
reaching values of 4 and 5 % for both the sun disk methods, not explainable at this state of the art. Both in Rome and
Valencia a generally better accordance with the laboratory calibration is for the solim3m method when in the range [400-
870] nm the difference is below 1.5% and 2.15% in Valencia and Rome, respectively. For the 1020 nm the comparison
in Rome has a larger difference up to 2.63%. Also for this POM a general overestimation of SVA from onsite calibration
is visible in Rome, as explained in the above paragraph.
Finally, we compared the performance of the on-site calibration procedure, method 3m, in Rome for the two co-located
instruments calibrated at the two different laboratories (Figure 14). The SVA values for POM_CNR better agree with the
calibration performed in AALTO laboratory, with the exception of 340 nm and 870 nm.

| | Wl [nm] | (%) diff s3m-s3n | (%) diff lab-3m | (%) diff lab-3n | | Wl [nm] | (%) diff s3m-s3n | (%) diff lab-3m | (%) diff lab-3n |
|---|---|---|---|---|---|---|---|---|---|
| | **AALTO calibration (lab)** | | | | | **PMOD calibration (lab)** | | | |
| POM_CNR ROME | 340 | -0.58 | 9.14 | 8.62 | POM_UV VALENCIA | 340 | -0.52 | -2.49 | -3.03 |
| | 400 | -0.56 | 0.03 | -0.53 | | 400 | -0.30 | -0.57 | -0.87 |
| | 500 | -0.46 | -1.95 | -2.41 | | 500 | -0.43 | -1.51 | -1.94 |
| | 675 | -0.51 | -1.18 | -1.70 | | 675 | -0.47 | -0.71 | -1.18 |
| | 870 | -0.95 | -3.46 | -4.45 | | 870 | -2.26 | 0.16 | -2.10 |
| | 940 | -1.38 | 1.12 | -0.25 | | 940 | -2.08 | -0.82 | -2.92 |
| | 1020 | -1.18 | -0.39 | -1.58 | | 1020 | -1.30 | -0.38 | -1.68 |
| POM_CNR IZANA | 340 | -0.27 | 10.92 | 10.68 | POM_UV ROME | 340 | -0.74 | -4.23 | -5.00 |
| | 400 | -0.34 | 1.10 | 0.76 | | 400 | -0.47 | -2.15 | -2.63 |
| | 500 | -0.28 | -0.95 | -1.23 | | 500 | -0.51 | -2.11 | -2.63 |
| | 675 | -0.41 | -0.34 | -0.75 | | 675 | -0.48 | -1.45 | -1.94 |
| | 870 | -0.47 | -2.07 | -2.56 | | 870 | -2.29 | -1.06 | -3.38 |
| | 940 | -1.21 | 1.73 | 0.54 | | 940 | -2.09 | -2.51 | -4.64 |
| | 1020 | -1.14 | 0.23 | -0.91 | | 1020 | -1.27 | -2.48 | -3.78 |

Table 6: Differences between SVA values from the onsite calibration methods and the laboratory calibrations for the two
POMs.



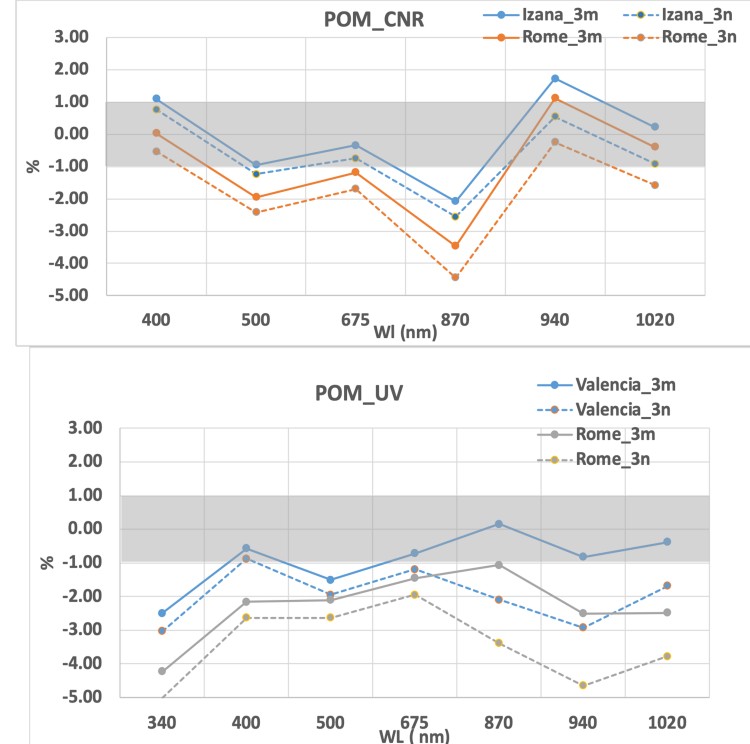


Figure 13: % difference of SVA values from sun disk methods and laboratory calibrations for POM_CNR (top) and
POM_UV (bottom).

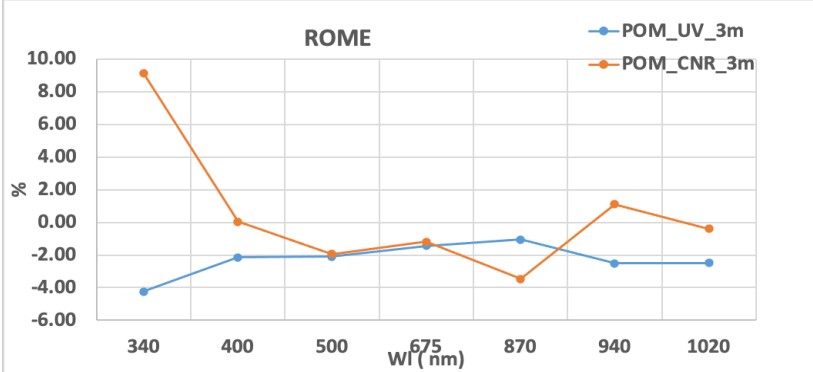

Figure 14: difference of SVA values from sun disk method 3m and laboratory calibrations for POM_CNR (orange) and
682                              POM_UV (blue) co-located in Rome.
**5. Conclusions**
The performance of the on-site calibration procedures applied to two PREDE-POMs instruments, was evaluated using
intercomparison campaigns and laboratory calibrations. Two periods were chosen for the validation: a) from September
2021 to November 2022, where 6 different calibration methodologies were compared against the SL method performed
in Izana in September 2022; the reference SL calibration was done in September 2022 and there is no availability of a
monthly reference calibration in the previous 12 months, to watch the stability of the instruments and check if their
shipments or usage affected the calibrations;  b) from August 2017 to September 2021, where the calibration transfer
from a PFR during the QUATRAM campaigns was used to evaluate the on-site methodologies.



The comparison against the SL showed an agreement generally improving with the wavelengths but with a small
worsening at 1020 nm. The IL works better in Davos with an agreement below 0.5% except at 1020 nm where it increases
up to about 1.5%. A very good accordance is also found in Valencia in November 2022, always within 0.8% except at
500 and 675 nm (within 1.5%). The similarity between the two cases is probably due to the very low turbidity recorded
in this month in Valencia, that makes the atmosphere optically more similar to the one in Davos. These results are in
accordance with Nakajima et al., (2021) where the estimation of the retrieval accuracy of $V_0$ from IL gives values of about
2.4% in Rome and around 0.3% - 0.5% at the mountain sites of Mt. Saraswati and Davos. These values are consistent
with the RMSD in the aerosol optical depth comparisons with other networks, that is less than 0.02 for $\lambda \geq 500$ nm and
about 0.03 for shorter wavelengths in city areas; smaller values of less than 0.01 are found in mountain comparisons.
The XIL provides a consistent improvement (with values within 1%) only in Rome for all the wavelengths, but in very
clean atmosphere as in Davos it was not possible to retrieve values at 1020 nm.
The 340 nm is the wavelength with the most problematic results for the on-site procedures in Rome (differences around
4%) probably because of the molecular polarization that causes calibration errors from IL and XIL methods at the UV
region (340 nm), especially in low aerosol loading atmosphere.
In Rome the calibrations transferred from PFR in September 2021 differ against the SL (performed in September 2022)
in the range [-2.1%; -1.9%] at 500 nm for the two POMs, and the difference with the transfer from Cimel is about -1.6%.
However simultaneous calculation of $V_0$ in September 2021 with IL and XIL at 500 nm provides values that differ from
the SL of less then 0.5% for POM_CNR and 1.2% from POM_UV. The reason of such discrepancy must be studied,
because is not attributable to a change in the equipment due to shipping or usage, since it would have been visible also
from the on-site methodologies.
For both the POMs the comparison with PTB laboratory calibration shows very high underestimations (down to -10%
except for POM_CNR, and -8% for POM_UV). The discrepancies between the laboratory-based values and the field
measurements are probably due to different operating conditions of the instruments (e.g., different alignment and
measurement geometries, operating modes, polarization, etc.) and unknown POM settings (e.g., POM temperatures,
signal readout procedures) under which the instruments were calibrated in the laboratory and used in the field.
The long term comparison of the on-site methods with the calibration transfer from PFR was performed in Davos and
Rome, and showed  for IL differences always greater than the uncertainties (%CV) of the method, for both wavelengths,
with the exception of Davos in 2017. Values are around 1% in Davos whereas the largest differences are in Rome and at
500 nm, likely due to the unfulfilled assumption that the refractive index do not largely change during the Langley plot.
O the other hand for XIL many differences are within the uncertainties (%CV) of the method, and those higher are closer
to the %CV values than in the IL method. XIL improves the agreement particularly in Rome where the largest difference
reduces from 3.5% to 2.5% at 500 nm and from 3% to 1.7% at 870 nm.
Future studies are planned to understand the effects of atmospheric scattering variability on the IL method and of the
molecular polarization on 340 nm, switching from the use of the Skyrad 4.2 pack to the Skyrad_MRI (Kudo et al.,2021).
A more close look at the effects on AOD is also underway on a second paper.

The solar disk scanning methods 3m and 3n performed in Rome and Izana were compared against the laboratory
calibrations. The difference varied from a minimum of 0.03% at 400 nm to a maximum of 3.46% at 870 nm in Rome and
from 0.23% at 1020 nm to 2.07% at 870 nm in Izana. Both the methods slightly overestimate the SVA values in Rome.
The 870 nm shows the highest discrepancy in both the sites and for both the solid3m and 3n methods for the two POMs.
A generally better accordance with the laboratory calibration was found for the solim3m method. An overestimation by
the on-site procedures in the range [500-870] nm wavelengths is observed in both the sites due probably to an effect of
the scattered light from aerosols and air molecules included in the measurement and to a contribution of the direct solar
light striking the lens. The scattering contribution and stray light reaching the detector increase the output, and the
integrated value has a larger magnitude that can affect the estimation of the SVA. The overestimation was lower in Izana
due to a less important scattering effect.
**Acronyms tables**

| ACTRIS | Aerosol, Clouds and Trace Gases Research Infrastructure |
|--------|--------------------------------------------------------|
| AM | Measured signal during solar disk scan |
| AOD | Aerosol Optical Depth |
| CIMO | Commission for Instruments and Methods of Observation |
| CV | Coefficient of Variation |
| DN | Digital Signals |
| DUT | Detector Under Test |
| DVM | Digital Voltage Meter |
| ERR | Errors |
| FOV | Field Of View |
| FRC | Filter Radiometer Comparison |
| FWHM | Full Width at Half Maximum |





| GAW | Global Atmospheric Watch |
|---|---|
| I/U | Current-To-Voltage Converter |
| IL | Improved Langley Method |
| LCD | Liquid Crystal Display |
| LSA | Laser Spectrum Analyzer |
| MAPP | Metrology for Aerosol optical Properties |
| MFRSR | Multifilter Rotating Shadowband Radiometer |
| MRI | Meteorological Research Institute |
| MUX | Multiplexer |
| NDF | Neutral-Density Filter |
| NIST | National Institute of Standards and Technology |
| OPO | Optical Parametric Oscillator |
| PFR | Precision Filter Radiometer |
| PMOD | Physikalisch-Meteorologische Observatorium Davos |
| POM_CNR | POM radiometer of Consiglio Nazionale delle Ricerche |
| POM_UV | POM radiometer of Univeristy of Valencia |
| PTB | Physikalisch-Technische Bundesanstalt laboratory |
| QUATRAM | QUAlity and TRaceabiliy of Atmospheric aerosol Measurements |
| REF | Reference |
| RMSD | Root Mean Square Deviation |
| SHG | Second Harmonic module |
| SL | Standard Langley method |
| STD | Standard Deviation |
| SVA | Solid View Angle |
| THG | Third Harmonic module |
| TULIP | TUable Lasers In Photometry |
| UV | Ultra Violet |
| VIS | Visible |
| WMO | World Meteorological Organization |
| WORCC | World optical depth research and calibration center |
| XIL | Cross Improved Langley method |
| ZM | calculated signals during the fitting phase in the solar disk scan |


**Competing interests**

At least one of the (co-)authors is a member of the editorial board of Atmospheric Measurement Techniques. Monica Campanelli and Stelios Kazadzis are co-editors of the Skynet Special issue.

**Acknowledgment**

This work has been supported by the European Metrology Program for Innovation and Research (EMPIR) within the joint research project EMPIR 19ENV04 MAPP. The EMPIR is jointly funded by the EMPIR participating countries within EURAMET and the European Union.

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
