# Peer review of "Evaluation of "on site" calibration procedures for sun-sky photometers"

_Atmospheric Measurement Techniques, 2023_

## Author Comment (AC2)

**Answers Ref 2**

The authors are grateful to the anonymous reviewer #2 for the careful reading of this work and for the many accurate suggestions and corrections.  In this response, we addressed each of the reviewer's points.

*The authors present an evaluation of on-site calibration procedures applied to two SkyNet Prede POM-01 sun/sky photometers.  The work is of good quality, and what has been presented has been done so thoroughly.  However, I believe major revisions are in order before this paper can be considered for publication.  I have identified gaps that need to be addressed, potential inaccuracies or misconceptions that should be addressed, found numerous issues with grammar and several typos, made constructive suggestions for organization that will improve clarity, and provided numerous recommendations to tables, equations, and figures in the attached PDF file.*

1) *The title should probably be changed to specify POM-01 sun-sky photometers since on-site procedures are ONLY applied to these instruments and not to any others.*

We accepted the suggestion of the reviewer, and the title now is: "Evaluation of "on site" calibration procedures for Skynet PREDE-POM sun-sky photometers". The method can be applied also to Cimels, as described in a previous paper (M. Campanelli, et al., 2007, DOI:10.1364/AO.46.002688) and we added this in the abstract and in the text stating that in this work, the method is only tested to PREDE POMs.

2) *This should say sun/sky photometers since classic sun photometers don't measure radiance….Intensive*

Done

3) *Prede (not capitalized) is the company name, not PREDE.  That is, it is not an acronym*

Done

4) *Specify POM-01 since Prede also produced POM-02 instruments which are not considered here.*

Done

5) *"The performance of the on-site calibration procedures for V0 was very good in sites with low turbidity, showing an agreement with a reference calibration between 0.5% and 1.5% depending on wavelengths. In the urban area, the agreement decreases between 1.7% and 2.5%. For the SVA the difference varied from a minimum of 0.03% to a maximum of 3.46%." Is it possible to relate these uncertainties to the retrieved quantities of AOD, SSA, and AAOD - even if only approximately?  Most users will not be familiar with the impact of a few percent uncertainty in SVA, specifically."*

These percentage values are not the uncertainties on the V0 or SVA evaluation, but they are the differences against reference or laboratory calibrations. A very approximate order of estimation of  AOD, SSA, and AAOD uncertainties, though the detailed accuracy of the retrieval also depends on the inversion program and a-priori aerosol climatology values as inversion constraint for the sun and sky data analysis, indicates that the 1% relative error of $V_0$ and SVA can cause respectively: 0.005 in AOD for measurements at the air mass of m=2, 0.01 in the SSA and 0.011 in AAOD. We added these comments in the Conclusion section, however the detailed effect on these uncertainties on the retrieved quantities of AOD, SSA, and AAOD is an investigation of another upcoming paper.

6) *"An error of 10% in the estimation of V0 induces an uncertainty in the retrieval of AOD of about 0.1," This is only true for an airmass of 1. For The AOD uncertainty scales as the inverse of airmass.*

Done

7) *"SVA is a measure of the field of view of the instrument," of the radiance measurement (which is not necessarily the same as the irradiance measurement in a general sense)*

Done

8) *"every six months operating at Mauna Loa and Izana and perform Langley calibrations" unclear. Rephrase please.*

Done

*9) English grammar makes this sentence ambiguous whether three PFRs are installed at each site (totalling 9 PFRs in all) or whether there is one at each of three locations. Consider "...reference PFRs are installed at..." where this implies one per listed locations.*

Done

*10) the word "rather" makes this ambiguous. Delete "rather".*

Done

*11) I recommend this be changed to "Instruments and Sites", and then proceed to describe the TWO POM systems that are the subject of the study. As it is, the instrument acronyms are used in Table 1 without first being defined or identified.*

*12) This creates an apparent conflict with the two Prede POM instruments that are the subject of this study. I STRONLY recommend that this section be entirely reworked to eliminate the first four QUATRAM campaigns that did not use either the POM_CNR or POM UV. This is all unnecessary information that promotes confusion on the part of the reader.*

*13) Recommend moving this phrase so that it immediate follows the specification of the seven filter wls*

As suggested by the reviewer we changed the title in "Instruments and site" section, and we re-wrote the section following this scheme:

i) Prede-POM description; the on-site calibration procedures, valuated in this work, were applied to four Prede-POMs (listed in Table 1). The subscript of the names of the POMs were modified and they are now described in the caption of Table I and in the Acronyms Table.
ii) Table I was improved by adding the instruments involved in each campaign
iii) PFR and CIMEL description.
iv) Sites and campaigns are described.
v) Explanation of the goal of the QUATRAM campaigns and their use in this study to evaluate the long-term differences between the on-site calibrations and the PFR transfer, as described in section 3.7 b.
vi) Explanation of the goal of the Izana and Valencia campaigns, held in the framework of the Metrology for aerosol optical properties (MAPP)

*14) STRONGLY recommend renaming this to POM-UVE (for University of Valencia , ES) to avoid confusion with Ultra-Violet.*

POM_UV was transformed to POM_VAL

*15) Consider adding a table listing all nominal wavelengths with columns with check-boxes for Cimel, Prede, and PRF channels. (Disregard the 315 nm Prede since it has been replace by 340 nm in both systems in this study, and exclude 940 nm because it requires use of a Langley approach modified for the non-linear absorption response of water vapor..*

All the wavelengths are well listed in all the other tables along the paper, and those that are compared are cleared explained also in the text. Therefore, we consider redundant the addition of a new table. The 940 nm was calibrated in laboratory for the calculation of the Solid View Angle and it important it is listed.

*16) mispelling: TUnable (also elsewhere in the text and in the list of acronyms)*

Done

*17) is this large difference in transmittance between the POM_UF and POM_CNR evident in other results? For example, does POM_UV 400 nm channel compare worse than POM-CNR 400 nm channel?*

No, it is not evident in other results. The comparisons between the two POMs are quite similar, with the exception of the Laboratory calibration and Standard Langley, that is better for POM_UV than for POM_CNR.

*18) In general three orders of magnitude are too much to show for uncertainties since uncertainties less than an order of magnitude than the largest will ultimately be negligible for consideration. Suggest removing the lines for u_current, u_trap, and maybe u_aperture which will allow a more appropriate vertical scale. (It's ok to let*

*u_wl be clipped as it drops below 1e-4; and for the same Figure: The legend labels are much too small for me to read without greatly expanding zoom. Please enlarge.*

Figure 3 shows spectrally dependent uncertainty components. As the reviewer correctly noted, components with values more than 3 orders of magnitude lower than the dominant ones have a negligible contribution. We have clipped the values at the lowest level of 1e-4 and removed u_current as this component can indeed be considered negligible. Nevertheless, u_trap and u_aperture remain in the figure. Their absence could be misinterpreted as providing incomplete uncertainty budget, even though they are typically among the dominant components in the uncertainty budget for calibration of laboratory equipment. The font size for the legend has also been increased.

19) *x = m_0 should go after "where"*

Done

20) *Major changes recommended for Tables 1 and 2. Recommendation #1: Remove columns for Vo. These numbers are really unique to the instruments since they rely on specific gain settings. They are meaningless for most users are are difficult to visually interpret since they have so many digits and also because variation with wavelength is partly due to the solar irradiance.*

The $V_0$ values doesn't show only values for several methods but allow to get difference among the calibration methods and/or against reference (i.e., Standard Langley). So, we believe important to show such "absolute" values. However, following the suggestion of the Reviewer we decided to move the Tables 2a, and b as they are in the appendix and to keep in the manuscript only the % values.

*Recommendation #2: Reduce the number of digits for %CV to just one decimal place. No one believes uncertainties to a hundreth of a percent.*

Reduced.

*Recommendation #3: Replace Unc with the percent relative uncertainty Unc/Vo. This will yield values in the same order of magnitude as %CV and will make your table much easier for the reader to digest.*

Probably there is a misunderstanding: UNC/V0 (in %) is exactly the % CV coefficient as explained in the paragraph before Eq 6.

*Recommendation #4: Do not intersperse Davos and Rome results. Instead, show all Davos results sorted chronologically and then all Rome results chronologically. This will group results with similar variability making it easier for readers to see trends.*

The order given in tables and plots is a "timeline". The instruments were submitted to continuous shipments, and a priori we don't know if they affected the equipment or not, therefore we would like to keep track of this in the sequence of V0 values. For example, one POM arrived at PTB laboratory with an internal electric board disconnected, and another one POM with the external optics completely dusty. A priori, the time pattern of V0 could depend also on these facts, however we are not able to discriminate them, as explained in section 3.7 before Eq. 10.

21) *In my view, this formulation of the improved Langley obsures the fact that it relies intrinsically on the retrieval process that incorporates the radiance measurement. I recommend making this very clear to the reader. In essence, the "improved Langley" attempts to account for the variation in AOD over the course of the Langley by using the fact that changes in AOD will also be reflected in changes in sky radiance. This is however complicated by the effects of multiple-scattering and SSA which is where the retrieval really becomes critcal. This where the problems arise in distinguishing changing AOD from errors in the retrieval process. Can you attempt to include some explanation of that in your text?*

Following the suggestion, and trying to better explain the method, we added the following paragraph in section 3.3: "As described in section 3.2, the standard Langley assumes that, in the selected time period, the AOD is constant, so data must be accurately chosen because the result is directly related to the variability of AOD. Shaw, 1979 and 1983, demonstrated that the linear dependence of AOD on $m_0$, which means a temporal change of the optical thickness because $m_0$ depends on time, corresponds to the second-order variation in terms of time. Limiting to the first order and following Eqs. 2 and 3 of Campanelli et al., (2004) the AOD can be expressed as the sum of a stable term ($AOD_0$) and a term indicating the variability ($AOD_1/m_0$). Eq.1 can be therefore briefly expressed as $lnV=lnV_0-AOD_1 -m_0 AOD_0$. In

the standard Langley plot the intercept value contains the variability (ln $V_0$ -AOD$_1$) and the retrieved $V_0$ value has a substantial dependence on the daily variability of AOD. Conversely in the Improved Langley plot $V_0$ is retrieved by the fit of ln$V$ versus the product of m$_0$ and the scattering optical depth that includes the variability term. In contrast to the standard method, the intercept $V_0$ does not depend on the AOD daily variation, if the product $_{ext}$ is correctly retrieved by the inversion process. To understand now the main idea on which this method is based,…."

22) $b_{IL} = -\frac{1}{\omega}$ .*Does this assume a constant SSA during the Langley? I think it does and this may not always be the case. This may be as serious an effect as changing the real index of refraction.*

In the present work we can only make an hypothesis about the reasons why in Rome the IL seems working not so good as in Davos, and we are studying the effect of the assumptions in a second paper. However we modified the comments about the results in Rome as it follows: "….even if, as shown in Nakajima et al. (2020), the IL accuracy is proportional to the optical thickness of the atmosphere of observation, generally low on high mountains. The same result has been also obtained by Ningombam et al. (2014). The greater differences are observed in Rome and at 500 nm. In this site AOD is higher than in Davos, as shown in Figure 6, and we would have expected a better performance of the on-site methodology. The reason of this result could be related to the fact that in the retrieval of $x$ for performing the fit in Eq.3, $\omega\tau_{ext}=\tau_{sca}$ and the refractive index must be assumed to not largely change during the Langley plot (Campanelli et al., 2004). In an urban site affected by traffic, as Rome, we can expect this assumption not satisfied. In this case the retrieved optical thickness can include an error caused by the inversion process for retrieving $\omega\tau_{ext}$ and also by an improper assumption of the refractive index."

23) *"In contrast to the standard Langley method, the intercept $V_0$ does not depend on the in-day variability of $\omega\tau_{ext}$ if the inversion process is accurate": This statement in inaccurate/incorrect. The standard Langley method is not sensitive to changes in SSA*AOD that vary day by day. It is susceptible to changes occuring _during_ the Langley retrieval. So long as AOD is constant over the duration of the Langley, the standard Langley is accurate. In contrast, the "improved Langley" is intended to be insensitive to variability within a given day, or more specifically during the Langley retrieval.*

"in-day" is the correct word, as the reviewer suggested, and it was changed.

*Just the real part or is this the complex refractive index. Please be explicit.*

The complex refractive index. We changed it in the text.

24) *Since you are looking directly at differences between IL and XIL approaches, it sure seems like the Cross IL method deserves its own heading and section.*

This paper is the first one publishing preliminary results on the XIL that, up to now, was only suggested in Nakajima et al., 2020). We still don't have detailed sensitivity studies, and we didn't apply it to other datasets, so we retain not having much material to do a separate section.

*Moreover, it would be instructive to show an example of a successful IL and its corresponding XIL Langley graphically.*

We added some examples in the appendix. However, every V$_0$ value from the IL or XIL method, is collected in a monthly series that is screened as described in section 3.3 and then averaged. A "successful IL", can be defined as a V$_0$ that agrees against a reference value (Section 3.7, a) but this value is not directly from a specific Langley but it is a monthly average of screened values.

25) *Cit AERONET*

Done

26) *Eq. 9 Express this equation to refer the reference system more generally and then eliminate Eq. which is redudantant.*

We acknowledge that Eq.9 is redundant, but we still retain it to facilitate the reading of this part of the paper keeping the subscripts that explain which instruments are analysed in each section. However, we modified Eq.7 in order to be in the same shape of Eq.9.

*27) replace this equation with one expressing the relative uncertainty which is much more useful in this context as you will see from tables 1 and 2 after they have been modified to include it.*

Eq 8 was modified.

*28) outside (This is a sublety of English usage. When we say "out of an interval" it really means "drawn out from within the interval". To indicate values not within the interval you need to say "outside".)*

Thank you for the kind explanation, we didn't know it. The word was changed.

*29) Move this comment closer to line 357 where the criterion was originally introduced.*

Done, the comment now is in point "i) data were selected between 9 to 13 UTC to avoid the rapid change in airmass"

*30) as is evident from Figure 4*

Figure 4 does not explain the frequent shipments, that conversely can be understood from Table I.

*31) "The agreement generally improves with the wavelengths but with a small worsening at 1020 nm." Can you conjecture why this might be the case? I actually dispute that it is true. I think your impression is being led astray by connecting the symbols with lines as a function of wavelength and by the dominance of the errors at 340 nm. If you disregard the results at 340 nm, the trend virtually disappears. I don't think you can say with confidence that there is a trend with wavelength or that 1020 nm is statistically worse than 870 nm. To prove this you'd need to conduct a statistical test.*

The reviewer is right. We deleted the sentence.

*32) "Also for the POM_UV, an improvement with the wavelengths is 440 notable with a worsening at 1020 nm." Again, I'm not sure this is true at all. I think your eyes may be deceiving you. Exclude 340 nm and the remaining points show scatter that is statistically flat with wavelength.*

We deleted also this sentence, in agreement with the above point.

*33) "CIMEL and especially POM have narrower field of views than the PFR, which makes them more susceptible to alignment and tracking errors, which could possibly lead to systematic underestimation of the measured irradiance values". However, in turbid conditions with low angstrom exponent forward scattering will tend to produce a high bias in instruments with wider FOV since they would be attributing forward scattered light to the direct beam irradiance. Unless you can say for certain that the Cimel and Prede are at the limit of their tracking accuracy I think you're going out on a limb in positing that they have an associated bias.*

In order to avoid misunderstanding and because of the large uncertainty in the explanation of the discrepancy against laboratory calibration, we decided to the delete the below paragraph, without trying to guess reasons.
"The instruments are obviously aligned and operated using different procedures when calibrated in the laboratory and when measuring in the field. CIMEL and especially POM have narrower field of views than the PFR, which makes them more susceptible to alignment and tracking errors, which could possibly lead to systematic underestimation of the measured irradiance values. It should be noted that the comparison results shown in Figure 4 are all from relative (Langley) measurements, with the exception of those based on the absolute responsivity calibrations at PTB, which makes the respective result in the comparison particularly sensitive to the effects mentioned above"

*34) This text gets lost between Tables 3 and 4. Recommend moving it below Table 4.*

Done

*35) Figure 5 Titles and x-axis labels are much too small for legibility.*

Fonts of Title and x-axis labels were increased.

*36) Figure 5 : Interspersing Davos and Rome results makes the results hard to interpret. Recommend significant changes to this figure. One option would be to make it a 4-panel plot with Davos results shown in the top row and Rome results in the botton (for example), or alternatively, keep it as a two panel plot but short the data first by site, and then by time. That is, group all the Davos results together, and then show all the Rome results.*

As explained in point 20, the order given in tables and plots is "chronological". Please refer to the reasons explained above.

 *"The largest differences are in Rome and at 500 nm, although the higher AOD as shown in Figure 6" I don't think this is a grammatical sentence. Please rephrase.*

The sentence was changed in "The greater differences are observed in Rome and at 500 nm. In this site the AOD is higher than in Davos, as shown in Figure 6, and we would have expected a better performance of the on-site methodology. The reason of this result could be…."

38) *Figure 9: This is not a single time series. It is really two. Plot the Davos and Rome data sets as their own time series, different color with different lines. This will help the reader see the trends for each site as well as the differences between the sites.*

Following the criteria of a temporal displacement of the instrument as justified in points 20 and 36, this Figure should keep the same order.

39) *"color aberration of the lens, diffraction at the edges": these show vary predictably as a function of wavelength*
*"misalignment of the optical axis": this should affect all channels equally*
*"surface nonuniformity of filters": this should be more or less random as a function of wavelength.*
*"sensor": This should be wavelength independent.*
*Whether the effects are wavelength dependent, constant, or random with wavelength are important considerations that you should mention and take into account in your discussion of the results.*

The sentence was changed as: "However, several factors contribute to this value: color aberration of the lens and misalignment of the optical axis, that are wavelength dependent, surface nonuniformity of filters randomly function of wavelength, and diffraction at the edges of the lens and non uniformity of the sensor that are wavelength independent."

40) *You could consider removing this section entirely. It adds nothing to the paper. It is just too noisy to be of much use.*

This paper is showing the results obtained from the MAPP-EMPIR project, and one of the objectives of this 3-years project, started in 2020, was the calibration of CIMELs, PREDE-POMs and PFRs, for radiance, irradiance, and field of view. Four laboratories were in-charge of these calibrations: AALTO, PMOD, PTB, VSL. For the POM-CNR and POM-VAL, two different methodologies to measure the FOV were tested by AALTO and PMOD respectively.. The purpose of section 4.1 is describing both methodologies and comparing their results against the solar disk scanning; therefore, we consider that section 4.1 is still an important contribution to this paper, also useful for researchers outside Europe to know the results of testing different methodologies

*Plus there are several unexplained details associated with the experimental layout. For example, is there a light baffle between the entrance and exit ports? Why is there a "water filter" in line with the Xe lamp?*

We added explanation of baffling in the figure caption of Fig. 8: "The integrating sphere is of coaxial type with a large screen between the entry and exit ports."

*What? A "water filter"? What purpose does it serve?*
The water filter removes heat at wavelengths above 1000 nm. We added this in Figure caption of Fig. 8. It is a baffle with running water inside to remove heat.

41) *Table 5. Recommend removing 940 nm throughout. It is not trivially Langley calibrated and will be subject to error sources in the retrieval process that muddy the issue.*

In Table 5, we are no more talking about solar calibration constant and Langley method, debated in section 3.7. Table 5 is part of section 4, where the field of view of the instruments is calculated using different methods at all the available wavelengths, included 940 nm that is used for the retrieval of columnar precipitable water content.

42) *- Figure 9: Labels, legend , red symbols are all to small"*

We increased the size of the symbols in the new Fig. 9.

*- Figure 9: what is "r" displayed in? Radians?! I hope not!*

"r" is the angular distance in degrees. We added the unit in Fig. 9. In the text it is written " The right figures present the signal intensity as a 1D function of distance (r) from the center of mass. But in the caption it is written r<0.19 degree. For clarification, we call r now "angular distance" in the text.

*- Figure 9: Why isn't a plateau evident in the measurements or traces?*

This is due to convolution. The setup mimics the geometry of viewing Sun and in such conditions, the field of view is of the same order of magnitude as the expectable plateau. We added a sentence on this in the text: "The measurements should form a plateau at small angles. However, this plateau is disturbed by convolution, as the resolution of the measurement is of the same order of magnitude as the plateau."

43) *Figure 10: Has the finite size of the source aperture diameter been taken into account? That is, did you de-convolve the cross-section of the apparent source from the observed FWHM. Doing so will increase your effective FOV by about 0.19 deg.*

Because the source apparent diameter of 0.19° is considerably smaller than the sun (apparent diameter 0.5°), which is the usual source that this instrument measures, the cross-section of the apparent source was not deconvolved from the measurements. Rather what has been done is to convolve the measurements with the apparent sun diameter to obtain the corresponding field of view. The slight error made by assuming an initial point source, instead of deconvolving the field of view, was assumed to be less than 0.5%, and added to the uncertainty budget. this explanation was added in the text.

44) *Is it elliptical to account for the fact that the azimuth angle steps are not actually equal to the elevation angle steps?*

The instrument automatically follows the sun during the scanning, lasting several minutes, and measurements are corrected for the movement of the solar disk, but to prevent the possible effect due to the difference between the azimuth and zenith angle steps, the system is considered elliptical. We added this comment in the test.

45) *" An elliptical system of coordinates centered at (0,0) is introduced and the needed parameters are obtained by fitting the measurements". This is a little short on detail.*

We added : "to prevent the effect due to the difference between the azimuth and zenith angle steps"

46) *Why is it called "solid3m"?*

The letter after 3 is the index of the version of the method.

47) *"The above-described method has been implemented by Uchiyama et al., (2018), (hereafter called solid3n) by not subtracting the minimum value largely affecting the measurements of the scattering angle between 1 and 1.4. and extrapolating the values between 1.4° and 2.5° using the data from 1.0° to 1.4°." I don't understand what is meant here "by not subtracting the minimum" etc. Consider rephrasing for clarity.*

The sentence was rephrased: "However, the subtraction of the minimum measured value largely affects the matrix of measurements in the range of scattering angles [1.0° to 1.4°], then Uchiyama et al., (2018) implemented the solid3m method, with a new version, hereafter called solid3n, that does not perform this subtraction, and extrapolates the values between 1.4° and 2.5° using the data from 1.0° to 1.4°."

48) *Just curious, why "solid3n"?*

According to my comment in point 45, n in the upgrade of version m.

49) *Figure 12: A few of suggestions here. 1) The blue colors are too similar. Consider removing 940 nm traces. That will allow 1020 nm to be green and more easily distinguishable. 2) Increase the size of all labels and legends. 3) Consider replacing the vertical axis with the percent variation about the mean. For POM UV It looks like almost +/- 5% which seems like a pretty large difference. What impact does this degree of uncertainty have on retrievals of aerosol intensive properties?*

Colours were changed. The SVA at 940 nm, as explained in point 41, was calibrated in the laboratory, therefore it is useful showing this result and compare it with the solar disk scanning method.

The coefficient of variation for the temporal variation (Std/mean) ranges from 1.1 to 1.3% for the POM_VAL (described in section 4.3). Hashimoto et al, 2012 (https://doi.org/10.5194/amt-5-2723-2012) demonstrated that a SVA underestimation of 1.4 % to 3.7 % can cause an increase of SSA of about 0.03 to 0.04. This estimation was done for Skyrad pack version 4.2. For the Skyrad_MRI_v2, actually used as Skynet standard inversion model, it is expected to be similar because the same forward model, RSTAR, is used in the retrieval, and the relation between SSA and diffuse radiance is the same. We added this information in the paper.

*50) "The solar disk scanning in Rome and Izana analyzed with the solid3m method agrees generally better, with respect to solid3n, with the laboratory calibration." Not grammatical. Rephrase please.*

The sentence was rephrased "The solar disk scanning matrixes, measured in Rome and Izana and analyzed with the solid3m method, provide SVA values that generally agree better with the laboratory calibration than those obtained by the solid3n."

*51) Recommend removing 940 nm throughout. It is not trivially Langley calibrated and will be subject to error sources in the retrieval process that muddy the issue.*
Please see my answer in point 41.

52) *Figure 13: As above, recommend eliminating 940 nm. It is an outlier, is not trivially calibrated with SL, IL, or XIL, and has error sources associated with changes in water vapor that don't show up in tau_sca.*
Please see my answer in point 41.

*53) Figure 14 Here again, remove 940 nm.*
Please see my answer in point 41.

*Note that if both 340 and 940 are excluded from the figure that there is really no observable trend with wavelength. In fact, 400, 870, and 1020 all show similar levels of variability. 500 and 675 show low variability but this may or may not be merely statistical.*

Yes, this is correct, but there is no reason to exclude the 340 and 940 wavelengths because both were calibrated in terms of solid view angles (not Langley's) by all the available methods and the 340 nm is also used for the retrieval of aerosol properties.

*54) "The comparison against the SL showed an agreement generally improving with the wavelengths but with a small worsening at 1020 nm." I disagree with this conclusion. If you really want to make this claim then you need to apply some statistical tests to confirm that trends are not merely random.*

In agreement with my answer in point 31, we modified this sentence "The comparison against the SL showed a very good agreement with many of the points within ±1%."

*55) "The 340 nm is the wavelength with the most problematic results for the on-site procedures in Rome (differences around 4%) probably because of the molecular polarization that causes calibration errors from IL and XIL methods at the UV region (340 nm), especially in low aerosol loading atmosphere." The problem with this supposition is that the polarization issue should be worse at low AOD, so should be worse for Davos than for Rome. Is it? If not, then this is "probably" not the explanation.*

The polarisation effect becomes, indeed, significant when AOD is low, therefore they should be more evident in Davos, but they also depend on the surface pressure (in Davos lower than in Rome) and therefore potentially weaker than in Rome. We added this sentence in the text.

*56) "Values are around 1% in Davos whereas the largest differences are in Rome and at 500 nm, likely due to the unfulfilled assumption that the refractive index do not largely change during the Langley plot." Does the retrieval assume a constant SSA during the Langley? This might also be an underlying issue. Or is that implied*

*in the refractive index statement? You should clarify whether you are referring to the real or complex refractive index.*

Please see answers in points 22 and 23

57) *"Measured signal during solar disk scan": Why is "AM" the acronym for "measured signal during solar disk scan"? Recommend changing this to another abbreviation or acronym since AM also means "before noon".*

The reviewer is right but AM and ZM are the original names of the variables for measured and calculate signals, inside the Fortran software, solid3m/n. We would prefer to keep the same names, so third users of the solid3m/n codes understand the text better .

58) *57 ) "DN: Digital signals" : Number. DN stands for Digital Number.*

Corrected

59) *Unfortunately, you are also using UV for University of Valencia. Recommend using UVE or UVES for University of Valencia, ES.*

See the answer in point 14.